Resource

# OrtSuite: from genomes to prediction of microbial interactions within targeted ecosystem processes

João Pedro Saraiva[1] , Alexandre Bartholomäus[2] , René Kallies[1] , Marta Gomes[3], Marcos Bicalho[1], Jonas Coelho Kasmanas[1,4,7], Carsten Vogt[1] , Antonis Chatzinotas[1,5,6], Peter Stadler[7,8,9,10,11] , Oscar Dias[3], Ulisses Nunes da Rocha[1]

**The high complexity found in microbial communities makes the identification of microbial interactions challenging. To address this challenge, we present OrtSuite, a flexible workflow to predict putative microbial interactions based on genomic content of microbial communities and targeted to specific ecosystem processes. The pipeline is composed of three user-friendly bash commands. OrtSuite combines ortholog clustering with genome annotation strategies limited to user-defined sets of functions allowing for hypothesis-driven data analysis such as assessing microbial interactions in specific ecosystems. OrtSuite matched, on average, 96% of experimentally verified KEGG orthologs involved in benzoate degradation in a known group of benzoate degraders. We evaluated the identification of putative synergistic species interactions using the sequenced genomes of an independent study that had previously proposed potential species interactions in benzoate degradation. OrtSuite is an easy-to-use workflow that allows for rapid functional annotation based on a user-curated database and can easily be extended to ecosystem processes where connections between genes and reactions are known. OrtSuite is an open-source software available at https://github.com/mdsufz/OrtSuite.**

## Introduction

In environments where microorganisms play a crucial role, the microbial community functional potential encompasses the building blocks for all possible interspecies interactions (Mulder et al, 2001; Maestre et al, 2012). For example, in environments rich in methane, microbial communities are dominated by species with genes encoding proteins involved in methanogenesis (Lyu et al, 2018). Soil microbes, especially those in the rhizosphere, are genetically adapted to support plants in the resistance against pathogens and tolerance to stress (Mendes et al, 2018). In this context, natural ecosystems are populated by an enormous number of microbes (Locey & Lennon, 2016). For example, soil environments can contain more than $10^{10}$ organisms per gram of soil heterogeneously distributed making a global search for interspecies interactions unfeasible (Raynaud & Nunan, 2014). The exponential increase in high-throughput sequencing data and the development of computational sciences and bioinformatics pipelines have advanced our understanding of microbial community composition and distribution in complex ecosystems (Roh et al, 2010). This knowledge increased our ability to reconstruct and functionally characterize genomes in complex communities, for example, by recovering metagenome-assembled genomes (MAGs) (Parks et al, 2017; Tully et al, 2018; Pasolli et al, 2019). Although several tools have been developed to improve the reconstruction of MAGs, the same cannot be said for predicting interspecies interactions (Morin et al, 2018). Studies by Parks et al (2017) and Tully et al (2018), although advancing the reconstruction of MAGs, did not perform any functional characterization or prediction of interspecies interactions. Pasolli and collaborators (Pasolli et al, 2019) performed functional annotation of representative species in their study by using several tools such as EggNOG (Huerta-Cepas et al, 2017), Kyoto Encyclopedia of Genes and Genomes (KEGG) (Kanehisa et al, 2004), and DIAMOND (Buchfink et al, 2015). However, the sheer number of representative genomes (4,930) and the lack of focus on specific ecosystem processes make predicting interspecies interactions challenging.

Furthermore, the challenge of predicting interspecies interactions increases because of the multitude of potential interactions between species in microbial communities and between microbes and their hosts (e.g., plants, animals, and microeukaryotes) (Slade

---

[1]Department of Environmental Microbiology, Helmholtz Centre for Environmental Research-UFZ, Leipzig, Germany   [2]GFZ German Research Centre for Geosciences, Section Geomicrobiology, Potsdam, Germany   [3]Centre of Biological Engineering, University of Minho, Braga, Portugal   [4]Institute of Mathematics and Computer Sciences, University of Sao Paulo, Sao Carlos, Brazil   [5]Institute of Biology, Leipzig University, Leipzig, Germany   [6]German Centre for Integrative Biodiversity Research (iDiv) Halle-Jena-Leipzig, Leipzig, Germany   [7]Department of Computer Science, Bioinformatics Group, Interdisciplinary Center for Bioinformatics, and Competence Center for Scalable Data Services and Solutions Dresden/Leipzig, University of Leipzig, Leipzig, Germany   [8]Max Planck Institute for Mathematics in the Sciences, Leipzig, Germany   [9]Institute for Theoretical Chemistry, University of Vienna, Wien, Austria   [10]Facultad de Ciencias, Universidad Nacional de Colombia, Bogotá, Colombia   [11]Santa Fe Institute, Santa Fe, NM, USA

Correspondence: ulisses.rocha@ufz.de

et al, 2017). An integrated pipeline for annotation and visualization of metagenomes (MetaErg) (Dong & Strous, 2019) attempted to address some of the challenges in metagenome annotation such as the inference of biological functions and integration of expression data. MetaErg performs comprehensive annotation and visualization of MAGs by integrating data from multiple sources such as Pfam (Mistry et al, 2021), KEGG (Kanehisa et al, 2004), and FOAM (Prestat et al, 2014). However, MetaErg's full genome annotation requires elevated processing times and computational resources due to its untargeted approach. In addition, there is a lack of a user-friendly tool to explore the results tables and graphs to extract pathway-specific information tied to each MAG and thus infer potential species interactions based on their functional profiles.

Genome-based modeling approaches have routinely been used to study single organisms as well as microbial communities (Gottstein et al, 2016). For example, constraint-based models are highly used in studying and predicting metabolic networks (Heirendt et al, 2019). These models are generated upon the premise that any given function is feasible as long as the protein-encoding gene is present. Although species may lack the genetic potential to perform all functions necessary to survive in a given ecosystem, outside laboratory conditions, microbes do not exist in isolation and may benefit from their interaction with other species. By assessing the genomic content of individual species, we are able to identify groups of microbes whose combined content may account for complete ecosystem functioning. However, generating full genome metabolic networks for each microbial community species is time-consuming as they require information not easily obtained for each community member, such as biomass composition and nutritional requirements.

To decrease complexity and facilitate analysis, it is possible to limit the search of interactions to groups of organisms (e.g., microbe–microbe or host–microbe) or specific ecosystem processes (e.g., nitrification or deadwood decomposition). A network-based tool for predicting metabolic capacities of microbial communities and interspecies interactions (NetMet) was recently developed (Tal et al, 2020). This tool only requires a list of species-specific enzyme identifiers and a list of compounds required for a given environment. However, besides the necessity of previous annotation of genomes, NetMet does not consider the rules that govern each reaction (e.g., protein complexes). Accurate annotation of gene function from sequencing data is essential to predict ecosystem processes potentially performed by microbial communities, particularly in cases where an ecosystem process is performed by the synergy of two or more species. Simple methods for the annotation of genomes rely, for instance, on the search for homologous sequences. Computational tools such as BLAST (Altschul et al, 1990) and DIAMOND (Buchfink et al, 2015) compare nucleotide or protein sequences to those present in databases. These approaches allow inferring the function of uncharacterized sequences from their homologous pairs whose function is already known. The degree of confidence in the assignment of biological function is increased if this has been validated by, for example, experimental data. Approaches based on orthology are increasingly used for genome-wide functional annotation (Huerta-Cepas et al, 2017). Orthologs are homologous sequences that descend from the same ancestor separated after a speciation event retaining the same function (Koonin, 2005). OrthoMCL (Li et al, 2003), CD-HIT (Li & Godzik, 2006), and OrthoFinder (Emms & Kelly, 2015, 2019) are just a few tools that identify

homologous relationships between sequences using orthology. OrthoFinder is more accurate than several other orthogroup inference methods because it considers gene length in detecting ortholog groups by introducing a score transformation step (Emms & Kelly, 2015). However, OrthoFinder, because of its all-versus-all sequence alignment approach, requires intensive computational resources resulting in long-running times when using large datasets for clustering. Because of the enormous number of potential combinations, limiting the scope of research to specific ecosystem processes may reduce the computational and resource costs associated with integrating ortholog clustering tools and functional annotation strategies. Still, having a pipeline that performs targeted annotation of genomes and genomic-based prediction of putative synergistic species interactions can assist researchers in the discovery of key players in any metabolic process. Furthermore, the identification of potential species interactions can lead to the design of synthetic microbial communities with a wide range of applications such as in bioremediation (Sharma & Shukla, 2020), energy production (Jiang et al, 2020) and human health (Clark et al, 2021).

In this study, we developed OrtSuite, a workflow that can (i) perform accurate ortholog-based functional annotation, (ii) reveal putative microbial synergistic interactions, and (iii) digest and present results for pathway and community driven biological questions. These different features can be achieved with the use of three bash commands in a reasonable computational time. This research question/hypothesis-targeted approach integrates a user-defined, Ortholog Reaction Association database (ORAdb) with up-to-date ortholog clustering tools. OrtSuite allows the search for putative microbial interactions by calculating the combined genomic potential of individual species in specific user-defined ecosystem processes. OrtSuite also provides a visual representation of the species' genetic potential mapped to each of the reactions defined by the user. We evaluate this workflow using a clearly defined set of reactions involved in the well-described benzoate-to-acetyl-CoA (BTA) conversion. Furthermore, we used this workflow to functionally characterize a set of known benzoate degraders. OrtSuite's ability to identify putative interspecies interactions was evaluated on species whose potential interactions have been previously predicted under controlled conditions (Fetzer et al, 2015).

## Results

One of the motivations to develop OrtSuite was to facilitate the targeted analysis of microbial communities' genomic potential, including the prediction of putative synergistic interspecies interactions. To simplify combining targeted functional annotation with the prediction of species interactions, we developed OrtSuite to integrate ortholog clustering tools (Emms & Kelly, 2019) with sequence alignment programs (Buchfink et al, 2015).

### OrtSuite is a flexible and user-friendly pipeline

Three simple-to-use scripts were created to collectively perform all tasks associated with OrtSuite and provide a user-friendly execution. Users would only be required to provide a list of identifiers

related to the ecosystem process of interest and the FASTA files (in protein format) of the species for which they intend to predict interactions. Next, the users only need to execute three simple bash commands that cover database generation, functional annotation, and species interactions.

Briefly, OrtSuite performs the following processing steps (Fig 1) (for further details see the Materials and Methods section). Step 1: In this step, the script *DB_construction.sh* takes the list of identifiers provided by the user and automatically downloads the protein sequences that will populate ORAdb. Step 2: In this step, the script *DB_construction.sh* takes the list of KO identifiers obtained during Step 1 and downloads the gene-protein-reaction (GPR) rules from KEGG modules. Step 3: In this step, the function *orthofinder* performs the clustering of orthologs. Step 4: In this step, the script *annotate_and_predict.sh* takes as input the FASTA files containing the ORFs of the genomes of interest and performs functional annotation (aligning them against the sequences in ORAdb). Step 5: In this step, the script *annotate_and_predict.sh* performs the prediction of putative synergistic interspecies interactions (Fig 1) using the output file "*Reactions_mapped_to_species.csv*" generated during Step 4. Although not necessary, additional control is given to the user with the option to establish thresholds in the minimum e-values (during sequence alignment of sequences in ortholog clusters to ORAdb). Other constraints include restricting the number of putative microbial interactions based on the presence of transporters and subsets of reactions to be performed by individual species (Table S1). Data in public repositories continue to be added or updated. Thus, manual inspection of the files in the ORAdb and GPR rules, although not mandatory, is strongly advised.

Users can choose from two alternatives to install OrtSuite. They may use a docker image for personal computers or conda packages (recommended for installation for High-Performance Computers). We created a user-friendly git repository (https://github.com/mdsufz/OrtSuite) that provides users with a user-friendly guide covering the installation and the three scripts used to run our pipeline and the generated outputs.

## Computing time of OrtSuite stages

We evaluated the runtime of each OrtSuite step on a set of genomes whose genomic potential in converting benzoate to acetyl-CoA was known (Table 1). OrtSuite was executed on a laptop with four cores

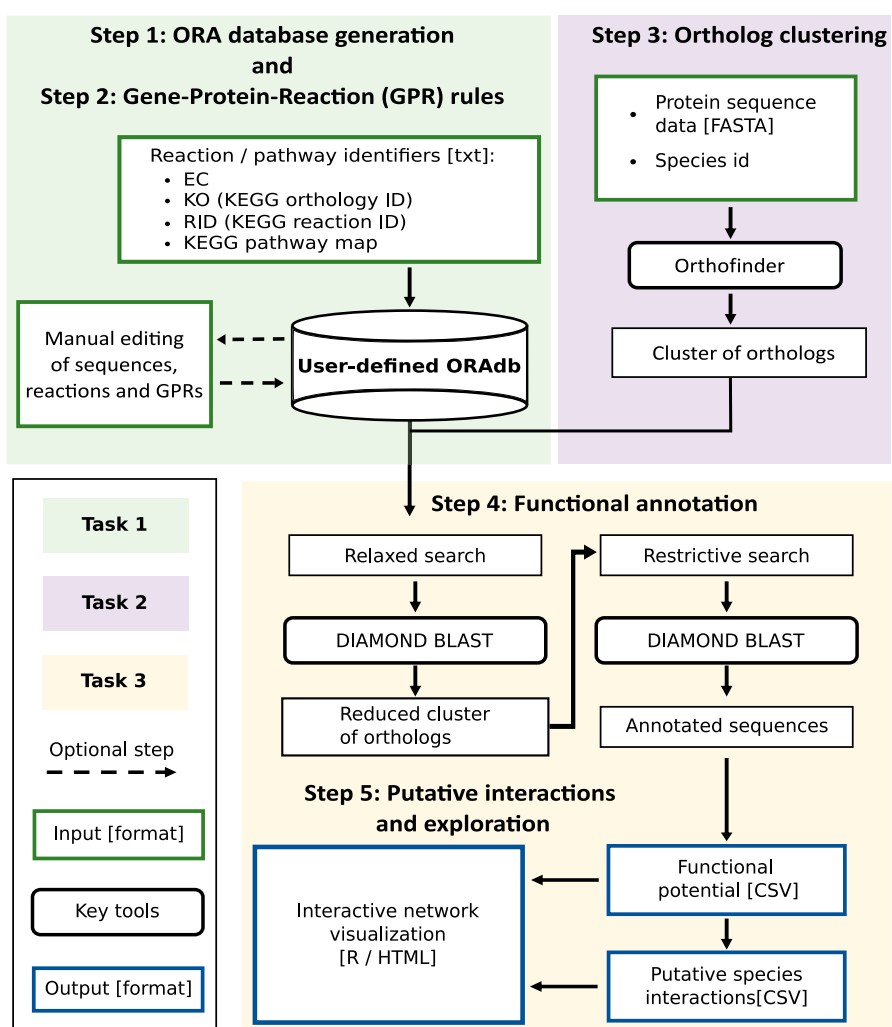

**Figure 1.  OrtSuite workflow.**
OrtSuite takes a text file containing a list of identifiers for each reaction in the pathway of interest supplied by the user to retrieve all protein sequences from KEGG Orthology and are stored in ORAdb. Subsequently, the same list of identifiers is used to obtain the gene-protein-reaction (GPR) rules from KEGG modules (Task 1). Protein sequences from samples supplied by the user are clustered using OrthoFinder (Task 2). In Task 3, the functional annotation, identification of putative synergistic species interactions and graphical visualization of the network are performed. The functional annotation consists of a two-stage process (relaxed and restrictive search). Relaxed search performs sequence alignments between 50% of randomly selected sequences from each generated cluster. Clusters whose representative sequences share a minimum E-value of 0.001 to sequences in the reaction set(s) in ORAdb continue to the restrictive search. Here, all sequences from the cluster are aligned to all sequences in the corresponding reaction set(s) to which they had a hit (default E-value = 1 × 10$^{-9}$). Next, the annotated sequences are further filtered to those with a bit score greater than 50 and are used to identify putative microbial interactions based on their functional potential. Constraints can also be added to reduce the search space of microbial interactions (e.g., subsets of reactions required to be performed by single species, transport-related reactions). In addition, an interactive network visualization of the results is produced and accessed via a HTML file.

**Table 1.   Species names, strain and abbreviation codes of the genomes used to validate OrtSuite (Supplementary data – Test_genome_set).**

| Name and strain | Abbreviation code | KEGG id | BTA pathway | Accession number | Reference |
|---|---|---|---|---|---|
| *Acinetobacter defluvii* WCHA30 | adv | T05474 | P3 | CP029389-CP029397 | Hu et al (2017) |
| *Arabidopsis thaliana* | ath | T00041 | — | GCF_000001735 | * |
| *Azoarcus sp.* KH32C | aza | T02502 | P2 | AP012304, AP012305 | Junghare et al (2015) |
| *Azoarcus sp.* DN11 | azd | T05691 | P2 | CP021731 | Devanadera et al (2019) |
| *Azoarcus sp.* CIB | azi | T04019 | P2 | CP011072 | Valderrama et al (2012) |
| *Burkholderia cepacia* DDS 7H-2 | bced | T03302 | P3 | CP007785-CP007787 | Jenul et al (2018) |
| *Burkholderia vietnamiensis* G4 | bvi | T00493 | P3 | CP000614-CP000621 | O'Sullivan et al (2007) |
| *Cycloclasticus sp.* P1 | cyq | T02265 | P3 | CP003230 | Wang et al (2008) |
| *Cycloclasticus zancles* 78-ME | cza | T02780 | P3 | CP005996 | Messina et al (2016) |
| *Desulfosporosinus orientis* DSM 765 | dor | T01675 | — | CP003108 | Robertson et al (2000) |
| *Aromatoleum aromaticum* EbN1 | eba | T00222 | P2 | CR555306-CR5553068 | Rabus et al (2016) |
| *Latimeria chalumnae* (coelacanth) | lcm | T02913 | — | GCF_000225785 | * |
| *Magnetospirillum sp.* XM-1 | magx | T04231 | P2 | LN997848-LN997849 | Meyer-Cifuentes et al (2017) |
| *Paraburkholderia aromaticivorans* BN5 | parb | T05169 | P3 | CP022989-CP022996 | Lee et al (2019) |
| *Rhodococcus ruber* P14 | rrz | T05142 | P3 | CP024315 | Peng et al (2018) |
| *Sulfuritalea hydrogenivorans* sk43H | shd | T03591 | P2 | AP012547 | Sperfeld et al (2019) |
| *Staphylococcus sciuri* FDAARGOS 285 | sscu | T05176 | — | CP022046-CP022047 | Mrozik and Labuzek (2002) |
| *Thauera sp.* MZ1T | tmz | T00804 | P2, P3 | CP001281-CP001282 | Suvorova and Gelfand (2019) |

The genomic potential, based on the KEGG database, to completely encode all proteins involved in a BTA pathway is identified in the column "BTA pathway" (P1: anaerobic conversion of benzoate to acetyl-CoA 1; P2: anaerobic conversion of benzoate to acetyl-CoA 2; P3: aerobic conversion of benzoate to acetyl-CoA). * indicates no literature was found connecting benzoate degradation and the respective species.

and 16 Gigabytes of RAM. We ran all OrtSuite steps on default settings and recorded the total runtime of each step (Table 2). The entire workflow was completed in 3 h 50 min, and the longest step was the construction of the ORAdb, which consisted of 2 h and 47 min which involved. The user can modify the number of cores used during functional annotation, further decreasing run times.

## Higher recall rates during clustering of orthologs with DIAMOND

Point mutations can have a drastic effect on the functional profile of microbes by altering the expected amino acid composition. Thus, we evaluated the impact of point mutations during the clustering of orthologs using OrthoFinder (Emms & Kelly, 2019). OrthoFinder allows users to choose between DIAMOND (Buchfink et al, 2015), BLAST (Altschul et al, 1990), and MMSeqs2 (Steinegger & Söding, 2017) as sequence aligners. DIAMOND and BLAST are the most commonly used sequence aligners. Therefore, we evaluated the clustering of orthologs these two tools. Nevertheless, the user may opt for MMSeqs2 as the sequence aligner when using OrtSuite. To test which of the two sequence aligners (DIAMOND or BLAST) yielded the best results, we performed ortholog clustering of a dataset consisting of the original target genomes and a set of artificially mutated genomes (Supplementary data - Test_genome_set) using both aligners. The results showed a 0.01 difference between BLAST and DIAMOND precision (Table 2). However, DIAMOND showed a 9.5% higher recall than that observed for BLAST what suggests DIAMOND may have higher sensitivity in the clustering of sequences with the same function. All artificially mutated sequences

(even those with mutation rates of 25%) were clustered together with their non-mutated ortholog. In parallel, we also performed sequence alignment using National Center for Biotechnology Information's (NCBI) BLASTp (Madden, 2002) between the protein sequences of the DNA-mutated and unmutated genes. e-values of sequence alignments in all species ranged from 0 to $5e^{-180}$ and percentage of identity from 61.32 to 98.84% (Table S2). For validation of the OrtSuite workflow, clustering of protein orthologs was repeated using only the original unmutated 18 genomes and the default aligner (DIAMOND). We also generated a complete overview of the results generated during the clustering of orthologs (e.g., number of genes in ortholog clusters, number of unassigned genes, and number of ortholog clusters) (Table S3).

## High rate of KEGG annotations predicted by OrtSuite

The third step of OrtSuite consists of performing cluster annotation in a two-stage process. In the first, only 50% of sequences are used to align the sequences from ORAdb. Those with a minimum e-value proceed to the second stage, where all sequences contained in this cluster will be aligned. At the end, annotation of clusters will take into consideration additional parameters such as bit scores. To evaluate the thresholds used in the annotation of ortholog clusters, we used one relaxed (0.001) and four restrictive ($1 \times 10^{-4}$, $1 \times 10^{-6}$, $1 \times 10^{-9}$ and $1 \times 10^{-16}$) e-value cutoffs. An overview of the results (e.g., number of clusters containing orthologs from ORAdb and number of ortholog clusters with annotated sequences) is shown in Table S4. The performance of OrtSuite in the functional annotation of the

**Table 2.   OrtSuite workflow runtime and clustering performance.**

| OrtSuite step | Runtime |
|---|---|
| ORAdb construction and Generation of GPR_rules | 2 h 47 min |
| Generation of protein ortholog clusters | 54 min |
| Functional annotation of sequences in ortholog clusters | 6 min |
| Defining putative microbial interactions | 3 min |
| Total | 3 h 50 min |
| Precision (BLAST) | 0.63 |
| Recall (BLAST) | 0.77 |
| Precision (DIAMOND) | 0.64 |
| Recall (DIAMOND) | 0.85 |

The total runtime of each OrtSuite step when analyzing the genomic potential of species in Test_genome_set dataset in three pathways (P1, P2, and P3) for the conversion of benzoate to acetyl-CoA (BTA). Steps were performed with default parameters on a laptop with four cores and 16 GB of RAM. Pair-wise precision and recall results of OrthoFinder using BLAST and DIAMOND as an alignment search tool. Clustering was performed on the Test_genome_set dataset plus the mutated genomes.

genomes in the *Test_genome_set* is shown in Table S5. On average, 96% of the annotations assigned by KEGG were also identified by OrtSuite. The complete list of functional annotation results using the different e-value cutoffs is available in the Tables S6–S9. Similarly, we used different e-value cutoffs for testing the mapping of species with the genetic potential for each reaction (considering the GPR rules) (Tables S10–S13). The four different e-value cutoffs used during the restrictive search stage yielded similar results in terms of annotation. However, the largest decrease in the number of ortholog clusters that transits from the relaxed search to the restrictive occurs when using an e-value cutoff of $1 \times 10^{-16}$ (Table S4). The difference in computing time between lower and higher e-value thresholds was negligible (<2 min). Other annotation tools, such as NCBI's BLAST tool (Altschul et al, 1990), BlastKOALA (Kanehisa et al, 2016), and Prokka (Seemann, 2014), can annotate full genomes, the latter at a relatively fast pace. On average, full genome annotation of our genomes in the *Test_genome_set* dataset using Prokka required 12 min on a standard laptop with 16 Gigabytes of RAM and four central processing units to complete. BlastKOALA required approximately 3 h to annotate a single genome. However, the use of these tools resulted in longer run times or in additional manual processing of files generated from full genome annotations for filtering pathways of interest.

### Identifying genetic potential to perform a pathway

To test OrtSuite's ability to identify species with the genetic potential to perform a pathway individually, we defined sets of reactions used in three alternative pathways for converting benzoate to acetyl-CoA (Table S14). Next, we compared the results to the species' known genomic content in each alternative pathway (Table S15). OrtSuite matched KEGG's predictions in species' ability to perform each alternative benzoate degradation pathway in all but two species, *Azoarcus* sp. DN11 and *Thauera* sp. MZ1T. Furthermore, OrtSuite identified five species capable of performing conversion

pathways not contemplated in KEGG. *Azoarcus sp.* KH32C, *Aromatoleum aromaticum* EbN1, *Magnetospirillum sp.* XM-1, and *Sulfuritalea hydrogenivorans* sk43H have the genetic potential to perform both pathways involving the anaerobic conversion of benzoate to acetyl-CoA, whereas *Azoarcus sp.* CIB has the genetic potential to achieve all alternative pathways (except when using an e-value cutoff of $1 \times 10^{-16}$). No genes in *Thauera sp.* MZ1T involved in the conversion of crotonyl-CoA to 3-hydroxybutanoyl-CoA (R03026) were identified by OrtSuite; this enzyme is essential for the anaerobic conversion of benzoate to acetyl-CoA. OrtSuite's performance yielded similar results between all tested e-value cutoffs. However, we observed a higher drop in the number of ortholog clusters whose sequences are all annotated with the same function when using an e-value cutoff of $1 \times 10^{-16}$. Thus, we set the default e-value for the restrictive search to $1 \times 10^{-9}$.

### Using OrtSuite to predict interspecies interactions

In this study, we tested the ability of OrtSuite in identifying interspecies interactions involved in the conversion of benzoate to acetyl-CoA where experimental data were available. We assessed the prediction of synergistic interspecies interactions on a set of sequenced isolates (Supplementary data - Fetzer_genome_set.zip). In a previous study (Fetzer et al, 2015), the potential of these isolates to grow in benzoate were analyzed individually and in combination under three different environmental conditions. These conditions were: low substrate concentration (1 g/l benzoate); high substrate concentration (6 g/l benzoate); and, high substrate concentration with additional osmotic stress (6 g/l benzoate supplemented with 15 g/l of NaCl). In that study, Fetzer et al (2015) investigated if the presence or absence of a particular species positively or negatively affected biomass production. Because under specific conditions, the presence of a degrader alone was not sufficient for biomass production, they further analyzed if potential species interactions could be of relevance. Briefly, Fetzer and collaborators defined minimal communities for all environmental conditions. Next, they tested whether the inclusion of other species in a community stimulated biomass production. When co-cultures produced biomass, the authors suggested the species in those communities had the potential to synergistically metabolize benzoate (Fetzer et al, 2015). Using OrtSuite, we aimed to identify which potential species interactions predicted by Fetzer and collaborators (Fetzer et al, 2015) could result from their combined genetic potential.

Our dataset contained 69,193 protein sequences distributed across the 12 species, resulting in 59 megabytes of data. More than 84% of all genes were placed in 9,533 ortholog clusters. In addition, 541 clusters were composed of sequences obtained from all 12 species (Table S16). OrtSuite's annotation stage resulted in 326 ortholog clusters with annotated sequences from ORAdb (Table S17). The mapping of KOs to each species in the *Fetzer_genome_set* is available as supplementary data (Table S18). The genomic potential of each species for aerobic and anaerobic benzoate metabolizing pathways is shown in Fig 2. The complete mapping of reactions to each species is available in the supplementary data (Table S19). Based on the 326 ortholog clusters and the GPR rules (Table S20), five species (*Cupriavidus necator* JMP 134, *Pseudomonas putida* ATCC 17514, *Rhodococcus sp.* Isolate UFZ (Umweltforschung

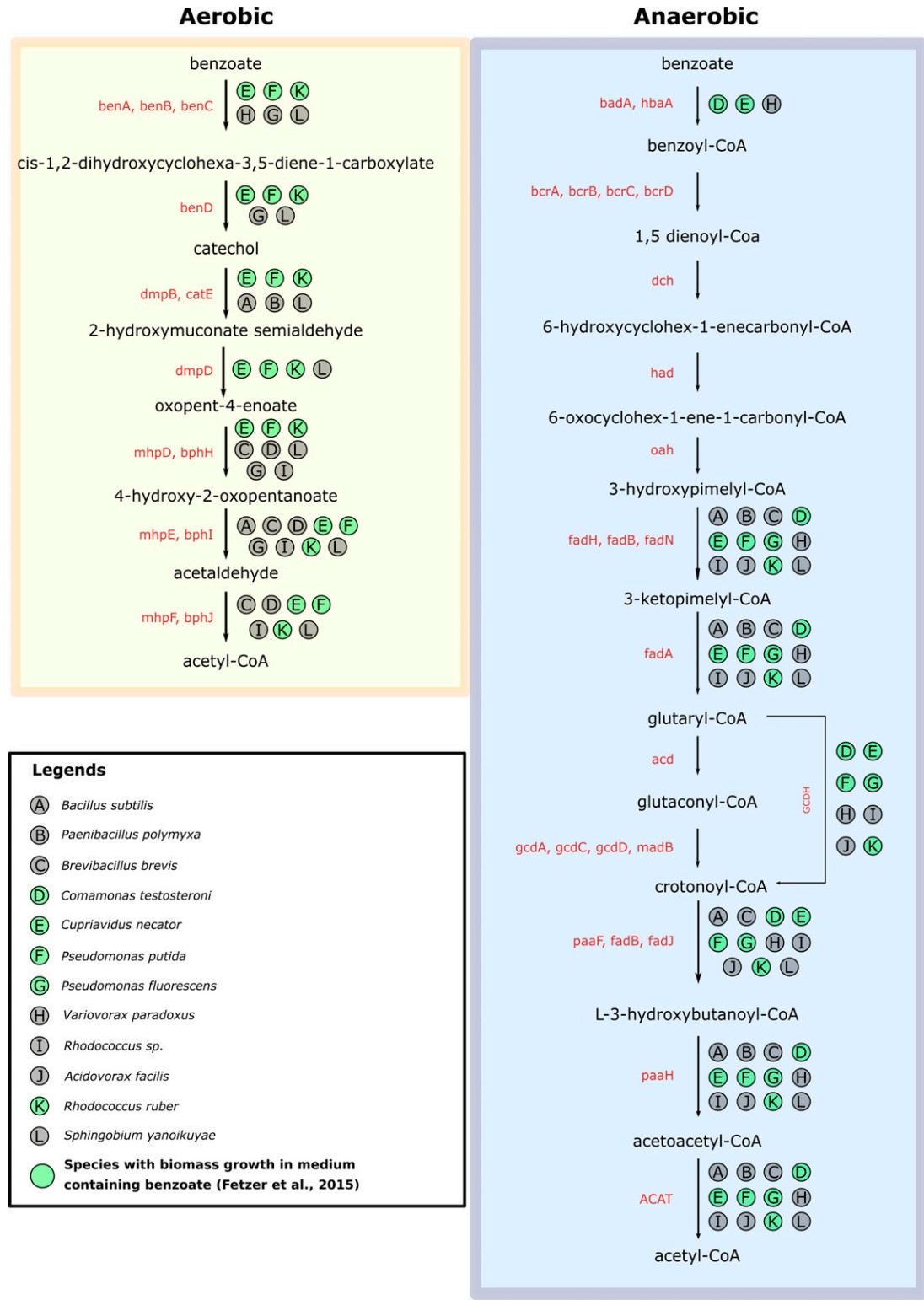

**Figure 2. Mapping of the Fetzer genome set to benzoate pathways.**
Mapping of the genomic potential of each species from the Fetzer_genome_set dataset to each reaction in aerobic (yellow) and anaerobic (blue) benzoate-to-acetyl-CoA conversion pathways. Circles highlighted in green represent species that showed biomass growth in medium containing benzoate in the Fetzer study.

Zentrum), *Rhodococcus ruber* BU3, and *Sphingobium yanoikuyae* DSM6900) contained all protein-encoding genes required to perform aerobic conversion of benzoate to acetyl-CoA. In the Fetzer study, *Rhodococcus sp.* Isolate UFZ and *S. yanoikuyae* did not show growth in a medium containing benzoate. The incomplete functional potential of *Comamonas testosteroni* ATCC 17713 and *P. putida* ATCC 17514 to perform aerobic conversion of benzoate to acetyl-CoA is at odds with their reported growth as monocultures in the presence of benzoate as shown in the Fetzer study. The number of species with the genetic potential for each reaction involved in the aerobic benzoate degradation pathway (P3) is shown in (Table S21). All species with the complete genomic potential to perform a complete pathway were excluded when calculating interspecies interactions because they do not require the presence of others. However, species identified by OrtSuite with the complete functional potential to perform each defined pathway were also included to compare to the results in the Fetzer study presented above. A total of 2,382 combinations of species interactions were obtained whose combined genetic potential covered all reactions. The complete list of potentially interacting species is available in the supplementary data (Table S22).

In the anaerobic degradation pathways (P1 and P2), no species presented the genomic content to encode proteins involved in the conversion of benzoyl-CoA to Cyclohexa-1,5-diene-1-carboxyl-CoA (R02451) (Table S23). This reaction requires the presence of a protein complex either composed of four subunits (K04112, K04113, K04114, and K04115) or composed of two subunits (K19515 and K19516). The genomes of the 12 species studied contained all subunits in either protein complex. Therefore, no species interactions were identified that would allow the complete anaerobic conversion of benzoate to acetyl-CoA. In the low substrate environment, OrtSuite identified 826 of 830 (99.5%) species combinations showing growth. In the high substrate environment, OrtSuite predicted 644 of 646 (99.7%). In the high substrate + salt stress environment, OrtSuite predicted all 271 (100%) combinations of species exhibiting growth (Table S24).

## Discussion

We designed OrtSuite to allow hypothesis-driven and user-friendly exploration of microbial interactions. Our team achieved this by integrating up-to-date clustering tools with faster sequence alignment methods and limiting the scope to user-defined ecosystem processes or metabolic functions. Using only three bash commands required to run the complete workflow, OrtSuite is a user-friendly tool capable of running in a customary computer (four cores and 16 GB of RAM) with even faster run times when using high-performance computing.

The clustering of orthologs by OrthoFinder using DIAMOND (Buchfink et al, 2015) showed higher sensitivity and lower runtime than BLAST (Altschul et al, 1990), which has also been shown by Hernández-Salmerón and Moreno-Hagelsieb (2020). Furthermore, low e-values and medium to high identity percentages in the sequence alignments between mutated and original genes indicated that the mutated genes still share enough sequence similarity to the original protein sequence. These results suggest that mutation rates of up to 25% of single DNA base pairs will not have an observable effect on the clustering of orthologs. OrthoFinder's algorithm removes the gene length bias from the sequence alignment

process, which may also explain why mutated genes were clustered with the original. Although it has been suggested that most genetic variations are neutral, changes in single base pairs can have a drastic effect on protein function (e.g., depending on the location of the mutation) (Ng & Henikoff, 2006). To this purpose, experimental functional studies can be used to validate previously unannotated orthologs. Furthermore, this study case does not consider the distribution of mutations across species and gene families, which can also have different effects on the clustering of orthologs (Khanal et al, 2015). Therefore, future studies increasing the rates of DNA base pair substitutions and other types of mutations and experiments targeting protein function in ortholog clusters are needed.

Next, we aimed to improve and facilitate functional annotation and prediction of synergistic microbial interactions. Exploring the great amount of data generated from full genome annotation of individual species from complex microbial communities is daunting. This is evident in a study by Singleton and collaborators (Singleton et al, 2021) where the connection between structure and function required the analysis of metagenomics data, 16S, and molecular techniques such as fluorescent in situ hybridization and Raman spectroscopy. When we looked at functional annotation alone, two challenges arose. First, performing all-versus-all sequence alignments in complex communities is resource-consuming (time and computational power). Second, manual inspection of each annotated genome for target genes or pathways is required. Identifying interspecies interactions based on the microbe's complete genomic potential is also challenging. For example, ecologists increasingly use network approaches, but selecting the most appropriate approach is not always straightforward and easy to implement (Delmas et al, 2019). OrtSuite overcomes these challenges by first performing cluster annotation in a two-stage process and limited to a user-defined set of functions, decreasing the number of sequence alignments necessary. The user-defined database coupled with the scripts for automated identification of interspecies interactions contained in OrtSuite decreases the time required to generate the data and facilitates its interpretation by the user. In addition, OrtSuite generates a graphical representation of the network enabling the use of the whole microbial community (Fig 3A–C) (https://github.com/mdsufz/OrtSuite/blob/master/network_example.png).

OrtSuite not only confirmed all but two of KEGG's predictions in species' ability to perform each alternative benzoate degradation pathway used in this study but also identified five species capable of performing conversion pathways not contemplated in KEGG. On average, an additional 18.3 KO identifiers were mapped to genes not previously annotated in the species used in this study. Using e-value and bit score as the filtering criteria rather than sequence identity, used by KEGG, may explain the increase in functionally annotated genes. For example, the alignment of a sequence of *Acinetobacter defluvii* (adv: AWL30228.1) to the sequences in ORAdb annotated as K04105 (conversion of benzoate to benzoyl-CoA) showed high bit scores (200.7) and low e-values ($2 \times 10^{-54}$) but the identity percentage did not exceed 28.6%. The use of e-values and bit scores to infer function has been reviewed by Pearson (2013). Pearson suggests that e-values and bit scores are more sensitive and reliable than identity percentages in finding homology because they consider the evolutionary distance of aligned sequences, the sequence lengths and the scoring matrix.

To test the prediction of putativeFigure synergistic microbial interactions, we used data from an independent study performed

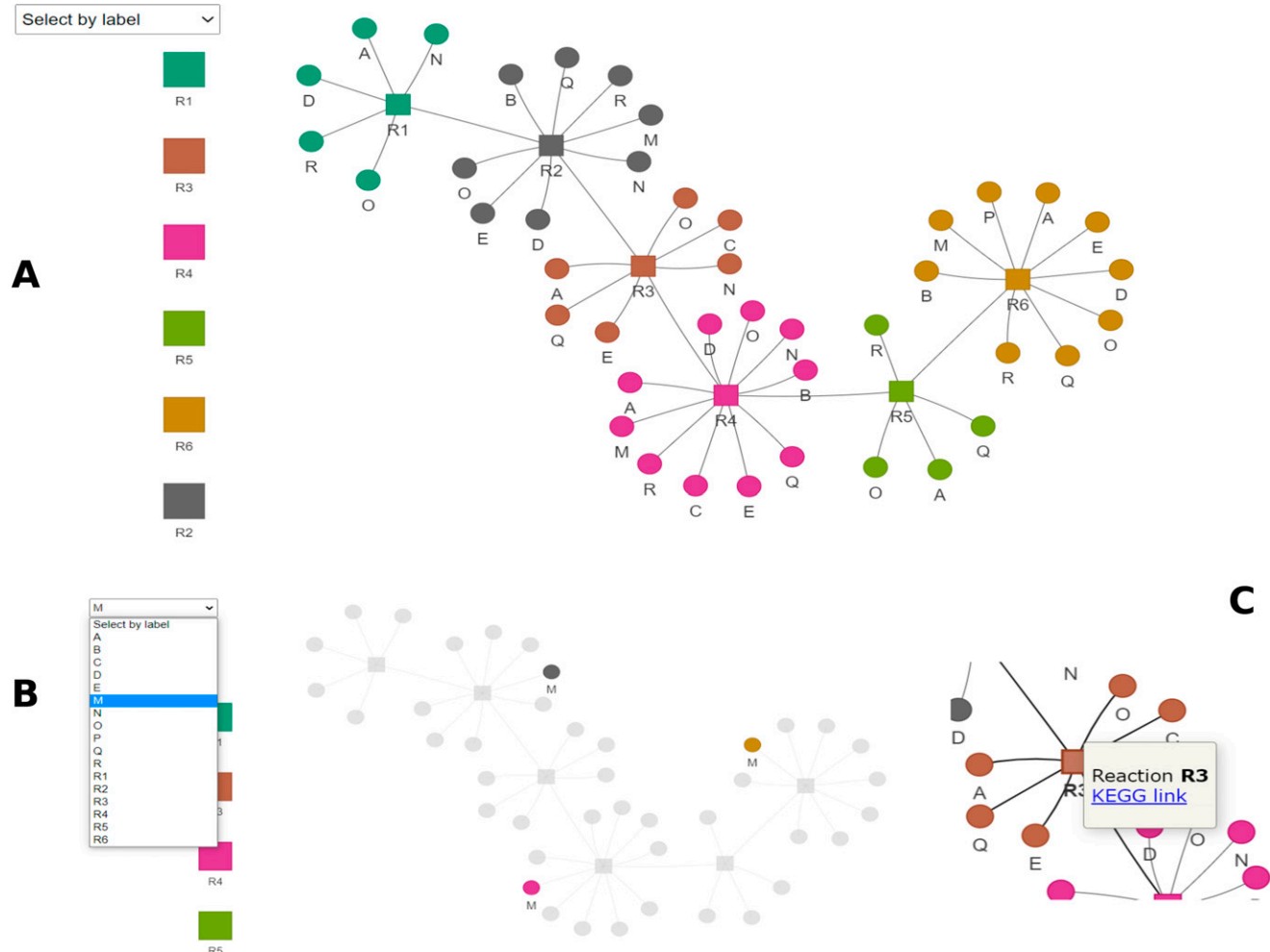

**Figure 3. Example of the interactive network visualization included on OrtSuite results.**
**(A)** The complete network with species is colored by reaction. **(B)** Species can be highlighted for simple identification. **(C)** Tooltips on reaction link out the KEGG if the reaction identifier is given.

by Fetzer and collaborators (Fetzer et al, 2015); hereafter, Fetzer study. In the Fetzer study, five species showed biomass growth (estimated by OD at 590 nm wavelength) in a medium containing benzoate. We evaluated whether these species possessed the complete genomic content to encode all proteins required for each benzoate to acetyl-CoA conversion pathway. The remaining seven species could not grow as monocultures in media with benzoate as the sole carbon source. Therefore, we evaluated whether the lack of growth was explained by the absence of essential protein-encoding genes involved in converting benzoate to acetyl-CoA. The Fetzer study also showed that, under specific nutrient and stress conditions, total biomass production was influenced by the presence of non-degrading species. Thus, we evaluated whether putative species interactions identified by OrtSuite fit the results obtained by in the Fetzer study. OrtSuite confirmed the functional potential for aerobic conversion of benzoate to acetyl-CoA in three of the five species whose growth in monocultures was observed during their research. In the Fetzer's study, *S. yanoikuyae* (accession number GCA_903797735.1) and *Rhodococcus sp.* (accession number

GCA_903819475.1) were not able to grow as monoculture in the presence of benzoate. However, OrtSuite predicted that both possessed the functional potential to aerobically convert benzoate to acetyl-CoA. In their study, growth was considered when OD were above 0.094. The OD measured for *S. yanoikuyae* was 0.0916 in a medium containing 1 g/l of benzoate. The annotation of genes with the ability to perform the complete aerobic conversion of benzoate to acetyl-CoA combined with a small difference in OD to the minimum threshold suggests that *S. yanoikuyae* can grow, albeit slowly, on low benzoate containing medium. In the case of *Rhodococcus sp.* Isolate UFZ, the OD was never measured above 0.022 what might indicate another slow-growing species. Another possible explanation is that although these two species possess the genes necessary for aerobic benzoate degradation, they are not active. In Fetzer's study, the observed growth of *C. testosteroni* ATCC 11996 and *Pseudomonas fluorescens* DSM 6290 in the low benzoate environment was not explained by OrtSuite. To note, benzoate conversion intermediates were not determined in the Fetzer experiment. Hence, these two species may use reactions or pathways

that were not included in the benzoate degradation pathways used in our study. Despite the presence of benzoate degraders, another possible explanation as to the unobserved growth in Fetzer's study for certain experimental conditions is the lack of tolerance of these species to high benzoate concentrations. For example, *C. necator* growth was stimulated at low benzoic acid concentrations but inhibited at high concentrations (Wang et al, 2014). In addition, the set of genes used in our study did not consider the presence of stress-related factors. To assess these effects, stress-resistance associated genes and reactions such as those involved in medium acidification (Kitko et al, 2009) could be added as constraints. Similar results were obtained when using a high substrate and salt stress medium. Under these conditions, the presence of benzoate degraders alone may not be sufficient to achieve growth. Benzoate degradation has been shown to decrease in hyperosmotic environments (Bazire et al, 2007). Therefore, additional constraints such as genes that confer resistance to environmental stressors or adverse conditions sodium chloride (NaCl) could be included in identifying interspecies interactions under different or changing environmental conditions.

No single species or combination of species possessed the complete genomic potential to anaerobically convert benzoate to acetyl-CoA via the two proposed pathways (P1 and P2). Because all growth experiments were conducted in aerobic conditions, the species in question may only use benzoate as a carbon source in aerobic environments. To fully explore all the species potential to convert benzoate, additional degradation pathways could be checked in the future using a multi-omics approach. For example, OrtSuite users could potentially integrate the use of (meta) transcriptomic data during the prediction of interspecies interactions by excluding species showing no gene expression of the selected pathways. However, the analysis and integration of (meta)transcriptomic data is not trivial and would add more levels of complexity to consider (e.g., expression of a gene can be high but protein be inactive) and is out of OrtSuite's scope. Furthermore, the only constraints added were related to the reactions that composed each pathway. Additional constraints can be included in future studies, such as potential mandatory transport-associated reactions, to increase confidence in the proposed interspecies interactions. Also, species interactions can be manually excluded if, for instance, antibiotic-producing species are known to inhibit the growth of others. OrtSuite confirmed that most interspecies interactions (>99%) identified by Fetzer and collaborators were possible because of their combined metabolic potential to aerobically degrade benzoate to acetyl-CoA but not under anoxic conditions.

In this study, we ran OrtSuite on a dataset composed of 18 genomes (Table 1). To determine if this range would be within the number of genomes in regular microbiome studies, we calculated the average number of MAGs from different studies focusing on their recovery. A study performed by Parks and collaborators (Parks et al, 2017) analyzed sequencing data from 149 projects. Most projects (91%) consisted of less than 20 samples. On average, they recovered 5.3 MAGs per metagenome. Work performed by Pasolli and collaborators (Pasolli et al, 2019) on microbial diversity in the human microbiome recovered, on average, 16 MAGs per metagenomic library. From the 46 studies used in their work, 30 consisted of less than 200 samples. Another study by Tully and collaborators focusing on marine environments (Tully et al, 2018) recovered 2,631 MAGs from 234 samples (average of 11 MAGs per sample). Our analysis demonstrates that the average number of MAGs recovered from a metagenomic library ranges from 5 to 16. Therefore, by using a regular

laptop, users can perform targeted functional annotation and interspecies interactions predictions using OrtSuite in average-sized metagenomes.

In summary, OrtSuite allows hypothesis-driven exploration of potential interactions between microbial genomes by limiting the search universe to a user-defined set of ecosystem processes. This is achieved by rapidly assessing the genetic potential of a microbial community for a given set of reactions considering the relationships between genes and proteins. The two-step annotation of clusters of orthologs with a personalized ORAdb decreases the overall number of sequence alignments that need to be computed. User-specified constraints, such as the presence of transporter genes, further reduce the search space for putative microbial interactions. Users have substantial control over several steps of OrtSuite: from manual curation of ORAdb, custom sequence similarity cutoffs to the addition of constraints for inference of putative microbial interactions. The reduction of the search space of synergistic interactions by OrtSuite will also allow more comprehensive and computationally demanding tasks to be performed. Such as (Community) Flux Balance Analysis, which depend heavily on genome-scale metabolic models (Thommes et al, 2019; Ravikrishnan & Raman, 2021 *Preprint*). As long as links between genes, proteins and reactions exist, the flexibility and easy usage of OrtSuite allow its application to the study of any given ecosystem process. Nevertheless, assessing the functional potential of microbes is just the first step in deciphering synergistic microbial interactions. Linking the functional potential of microbial communities to transcriptomic or proteomic data will improve predictions and provide further insights into other types of microbial interactions.

# Materials and Methods

### OrtSuite workflow

The OrtSuite workflow consists of three main tasks performed using three bash commands (Fig 1). The first task consists of generating a user-defined Ortholog Reaction Association database (ORAdb) and collecting the GPR rules. This task takes as input a list of KEGG identifiers which will be used to download all protein sequences associated with a set of reactions/pathway of interest. Next, all GPR rules associated with each reaction will be downloaded from KEGG Modules. In the second task, OrtSuite uses OrthoFinder (Emms & Kelly, 2015) to generate ortholog clusters. This task takes as input a folder with the location of the genomic sequences. The third task consists of the functional annotation of species, identification of putative synergistic interspecies interactions, and generation of visual representations of the results.

### OrtSuite task 1 (green box, Fig 1): user-defined ortholog-reaction association database (ORAdb) and GPR rules file

The ORAdb used for functional annotation consists of sets of protein sequences involved in the enzymatic reactions that compose a pathway/function of interest defined by the user. This database is generated during the execution of the *DB_construction.sh* script in OrtSuite, requiring the user to provide:

● A location of the project folder where all results will be stored

- A text file with a list of KEGG identifiers (one identifier per line)
- The full path to the OrtSuite installation folder

The list of identifiers can be KEGG reactions (RID) (e.g., R11353, R02451), enzyme commission numbers (e.g., 1.3.7.8, 4.1.1.103) or KEGG ortholog identifiers (e.g., K07539 and K20941). This file is used by OrtSuite to automatically retrieve the KEGG Ortholog identifiers (KO) (in case the identifiers provided are not KO identifiers) and to download all their associated protein sequences (Kanehisa et al, 2004). OrtSuite makes use of the python library *grequests*, which allows multiple queries in KEGG and subsequently decreases the time required for retrieving the ortholog associated sequences. The user-defined ORAdb will be composed of KO-specific sequence files in FASTA format associated with all reactions/enzymes of interest. Users also can manually add or edit the sets of reactions and the associated protein sequences in the ORAdb. This feature is particularly important because many reactions related to ecosystem processes are constantly being discovered and updated and might not be included in the latest version of KEGG. In addition, during the execution of the D*B_construction.sh* OrtSuite performs the automated download of the GPR rules from KEGG Modules. This feature is vital because enzymes can catalyze many reactions with a single (i.e., one protein) or multiple subunits (i.e., protein complexes). We advise users to manually curate the final table to guarantee accurate results despite the automated process. An example of the final GPR table is shown in the Supplementary data (Table S20).

**OrtSuite task 2 (purple box, Fig 1): generation of protein ortholog clusters**

The second task of OrtSuite, takes a set of protein sequences and generates clusters of orthologs. This set of protein sequences can originate from single isolates or from the complete set of protein sequences recovered from metagenomes or MAGs. Indeed, using protein sequences from isolates, MAGs, and co-culture experiments will benefit significantly from OrtSuite's reduction of the universe of potential microbial interactions based on the user-defined ORAdb. Orthology considers that phylogenetically distinct species can share functional similarities based on a common ancestor (Gabaldón & Koonin, 2013). Potentially, genes with similar functions will be grouped together. To perform this task, the OrtSuite pipeline uses OrthoFinder (Emms & Kelly, 2015). Three sequence aligners are available in OrthoFinder–DIAMOND (Buchfink et al, 2015), BLAST (Altschul et al, 1990), and MMSeqs2 (Steinegger & Söding, 2017). DIAMOND (v0.9.22) is used by default because of its improved trade-off between execution time and sensitivity (Emms & Kelly, 2019). This task is performed by running the command *orthofinder* located in the installation folder of OrthoFinder. This command takes as input the full path to the folder containing the protein sequences to be clustered and the full path to the folder where results are to be stored.

**OrtSuite task 3 (yellow box, Fig 1): functional annotation of ortholog clusters**

The third task of OrtSuite consists of the assignment of functions to protein sequences contained in the ortholog clusters. Functional annotation of these clusters consists of a two-step process termed

relaxed and restrictive search, respectively. The goal of the relaxed search is to decrease the number of alignments required to assign functions to sequences in the ortholog clusters. Here, 50% of the sequences from each cluster are randomly selected and aligned to all sequences associated with each reaction present in the ORAdb. Only the e-value is considered during this stage. Ortholog clusters where e-values meet a user-defined threshold to sequences in the ORAdb proceed to the restrictive search. The default e-value was set to 0.001, as the main objective of the relaxed search is to capture as many sequences for annotation as possible while avoiding an excessive number of sequence alignments. In the restrictive search, all sequences in the transitioned ortholog clusters are aligned to all the sequences in the reaction set(s) present in the ORAdb to which they had a hit during the relaxed search. Again, the query sequence is only assigned to the function of a reference sequence if the e-value is below a determined threshold (default $1 \times 10^{-9}$). Next, an additional filter is applied based on annotation bit score values (default 50). Although we established default values for the relaxed and restrictive search and bit score, the user can define the thresholds for all individual parameters.

The identification of putative interactions between species is based on all combinations of bacterial isolates with the genomic content to perform the user-defined pathway defined in the ORAdb. The input for this task consists of: (1) a binary table generated at the end of the functional annotation, which indicates the presence or absence of sequences annotated to each reaction in the ORAdb in each species (e.g., Table S10); (2) a set of GPR rules for all reactions considered (e.g., Table S20); and (3) a user-defined tab-delimited file where the sets of reactions for complete pathways, subsets of reactions required to be performed by single species and transporter-associated genes (e.g., Table S1) are described. Manual filtering can be performed to further reduce the vast amount of putative microbial interactions and increase confidence in the results. For example, results can be queried for known cross-feeding relationships between species or interactions that remove toxic compounds. Also, putative interactions can be removed if they are not biologically feasible. The user also may have an interest in assessing subsets of microbial interactions using specific criteria. Therefore, additional constraints can be applied to the putative microbial interactions, further reducing the search space. These include the degree of completeness of a pathway, the number of reactions expected to be performed by a single species or the presence or absence of transporter genes. In addition, graphical network visualization is also produced during this step (Fig 3A–C). The graphical network visualization is implemented in R using the packages visNetwork (v2.0.9), reshape2 (v1.4.3), and RColorBrewers (v1.1-2) but also requires the pandoc linux library. Graphical visualization was implemented with R v3.6 but also tested with v4.0. The visualization creates a HTML file that allows interactive network exploration and provides hyperlinks to KEGG if available.

All tasks—functional annotation, prediction of putative microbial interactions, and generation of graphical visualizations—are performed by running the script *annotate_and_predict.sh* included in OrtSuite (https://github.com/mdsufz/OrtSuite/blob/master/annotate_and_predict.sh). OrtSuite's predictions of individual species and combinations of species with the genetic potential to perform each defined pathway are stored in text files located in a folder termed "interactions."

## Conversion of benzoate to acetyl-CoA as a model pathway

We selected three alternative pathways involved in the conversion of benzoate to acetyl-CoA (BTA) to test the functional annotation and prediction of putative synergistic microbial interactions using OrtSuite (Table S14). Two pathways consisted of benzoate's anaerobic degradation to acetyl-CoA via benzoyl-CoA differing only in the reactions required for transformation of glutaryl-CoA to crotonyl-CoA (hereafter, respectively, P1 and P2). P1 first converts glutaryl-CoA to glutaconyl-CoA and then to crotonoyl-CoA, whereas P2 directly converts glutaryl-CoA to crotonoyl-CoA. One pathway consisted in the aerobic degradation of benzoate via catechol (hereafter P3). The complete number of reactions, enzymes, KO identifiers and KO-associated sequences in each alternative pathway is shown in the supplementary data (Table S25).

## Species selection for testing functional annotation

To assess the performance of OrtSuite, we selected the transformation of benzoate to acetyl-CoA as a model pathway and a set of previously characterized species known to be involved in this pathway (Table 1). This set of species was divided in two groups. The first group contained sequenced genomes of species whose ability to convert benzoate to acetyl-CoA has been demonstrated by KEGG (Kanehisa et al, 2004) and were selected as positive controls. These species were classified according to their genomic potential: complete, if all protein-encoding genes required for a BTA pathway were present in their genome or partial, if not all protein-encoding genes were present. The second group consisted of species that lacked all required protein-encoding genes and were selected as negative controls. In total, we selected 18 species as positive controls. Seven of them have the genetic potential to perform the alternative P2 pathway, eight have the genetic potential to perform alternative path P3 (positive controls), and none can completely perform the alternative path P1. To note that species *Thauera sp.* MZ1T has the genetic potential to perform P2 and P3 pathways. Four organisms were selected as negative controls. Using their genomes, we evaluated the performance of OrtSuite based on precision and recall rates for clustering of orthologs and the correct functional annotation of sequences. Also, a set of genomes from the species containing the genetic potential to degrade benzoate (*Burkholderia vietnamiensis* G4, *Azoarcus sp.* CIB and *Aromatoleum aromaticum* EbN1) were artificially mutated at the nucleotide level at different rates to determine how levels of point mutations in ORFs affected clustering of ortholog groups.

## Species selection for validation of putative interspecies interactions

In a study performed by Fetzer and collaborators (Fetzer et al, 2015), community biomass production of mono- and mixed-cultures was assessed in a medium containing benzoate. The authors used these data to infer potential species interactions. We processed this set of genomes with OrtSuite to determine the species' genetic potential to degrade benzoate, either individually or because of their interaction. Our results were compared with those obtained by Fetzer and collaborators and used to assess whether potential microbial interactions could be derived from their combined genetic potential.

## Evaluation of ortholog clustering

We evaluated the clustering of orthologs by measuring the pair-wise precision and recall. Clustering precision measures how many pairs of sequences associated with the same molecular function are grouped and is calculated by dividing the number of correctly clustered sequences by the total number of clustered sequences (Equation (1)).

$$\text{Clustering precision} = \text{correctly clustered sequences}/ \text{total number of clustered sequences,} \tag{1}$$

where correctly clustered sequences refer to the pairs of sequences that share the same function and are clustered together and total number of clustered sequences refers to all pairs of sequences that are clustered together irrespective of sharing the same function.

Clustering recall measures how many pairs of sequences with the same molecular function are not clustered together. Recall is calculated by dividing the number of correctly clustered sequences by the total true sequence clusters (Equation (2)).

$$\text{Clustering recall} = \text{correctly clustered sequences}/ \text{total true sequence clusters,} \tag{2}$$

where correctly clustered sequences refer to the pairs of sequences that share the same function and are clustered together and total true sequence clusters refers to all pairs of sequences that have the same function.

## Evaluation of sequence aligner used for clustering of orthologs

Changes of a single DNA base can produce a different amino acid, which might result in a different protein. To determine the impact of mutations on the clustering of orthologs a single gene from three species was artificially mutated at different rates. These mutations were introduced in the nucleotide sequences of each gene. Only substitutions were considered because these are the most commonly studied (Lynch, 2010), and none of the mutations were allowed to occur on the first and last codon. When, during the mutation, new stop or/and start codons were introduced, the translation was made for all the possible proteins and the largest was selected.

*Burkholderia vietnamiensis* G4 was mutated on the gene K05783, *Azoarcus sp.* CIB on the gene K07537 and *Aromatoleum aromaticum* EbN1 on the gene K07538. Each gene was mutated at rates of 0.01, 0.03, 0.05, 0.1, 0.15, and 0.25. Each mutation rate resulted in an in silico strain of the original genome (e.g., *Burkholderia vietnamiensis* G4 strain K05783_25, where "K05783" is the KEGG ortholog identifier and "25" is the rate of mutation). A total of 18 strains were generated (six in silico mutated strains per genome). The complete set of original and artificially mutated genomes is available in a compressed file (Supplementary data - Test_genomes_set.zip).

## Evaluation of functional annotation

Functional annotation was evaluated based on the data collected from KEGG (Altschul et al, 1990). Annotation performance is calculated

by dividing the number of matching annotated sequences by the total number of annotations (Equation (3)).

$$\text{Annotation performance} = \text{matching annotated sequences}/ \text{total number of annotations}, \tag{3}$$

where matching annotated sequences refers to the number of sequences annotated by KEGG annotations predicted by OrtSuite and total number of annotations refers to the all sequences that were assigned a function by KEGG.

### Evaluation of microbial interaction predictions

We evaluated the prediction of putative microbial interactions using a genome set from an independent study (Fetzer et al, 2015) containing species with exhibited growth in medium containing benzoate (defined as Fetzer_genome_set). The authors do not identify specific potential interactions in the transformation of benzoate but infer interspecific interactions in an environment containing benzoate as the major carbon source. For the complete set of species combinations and benzoate degradation capabilities and effects identified by Fetzer and collaborators, see Fetzer et al (2015) (Table S24).

#### *Bacterial cultures and sequencing*
Bacterial cryo-cultures of the different isolates were revived on LB agar plates. Single colonies were picked and grown overnight in 2 ml LB medium at 37°C. The cells were pelleted by centrifugation. Cells were lysed and genomic DNA was extracted using a Nucleospin Tissue Kit (Machery and Nagel). Approximately 150–1,000 ng of DNA were used for fragmentation (insert size: 300–700 bp) and sequencing libraries were prepared following the NEB Ultra II FS Kit protocol (New England Biolabs). Libraries were quantified using a JetSeq Library Quantification Lo-ROX Kit (Bioline) and quality-checked by Bioanalyzer (Agilent). These libraries were sequenced on an Illumina MiSeq instrument with a final concentration of 8 PM using the v3 600 cycles chemistry and 5% PhiX.

#### *Genome assembly and open reading frame prediction*
The sequenced reads were quality-checked using Trim Galore v0.4.4_dev. Next, genomes were assembled using the Spades Assembler v3.15.2 and their quality assessed using CheckM. Taxonomic classification was performed using Genome Taxonomy Database (GTDBTk) release 95. ORFs were predicted using Prodigal v2.6.3. Translation of sequences to amino acid format was performed using faTrans from kentUtils (https://github.com/ENCODE-DCC/kentUtils/tree/master/src/utils/faTrans).

## Data Availability

The datasets and computer code produced in this study are available in the following databases:

- The genomes used to test the workflow are available at National Centre for Biotechnology Information (https://www.ncbi.nlm.nih.gov/) under the accession identifiers CP029389-CP029397, GCF_000001735, AP012304, AP012305, CP021731, CP011072, CP007785-CP007787, CP000614-CP000621, CP003230, CP005996, CP003108, CR555306-CR5553068, GCF_000225785, LN997848-LN997849, CP022989-CP022996, CP024315, AP012547, CP022046-CP022047, and CP001281-CP001282.
- The genome assemblies used to predict interspecies interactions are available at National Centre for Biotechnology Information (https://www.ncbi.nlm.nih.gov/) with the study accession PRJEB38476: (https://www.ncbi.nlm.nih.gov/bioproject/648592).
- OrtSuite scripts: GitHub (https://github.com/mdsufz/OrtSuite).

## Supplementary Information

## Acknowledgements

We thank the early users of OrtSuite Sandra Silva, Felipe Côrrea, for their help with debugging and for workflow suggestions. We also thank Diogo Lima and Emanuel Cunha for their assistance in the implementation of the script required to generate the gene-protein-reaction rules, and Nicole Steinbach for her work in the sequencing of the isolates used as the Fetzer test set. This work was funded by the Helmholtz Young Investigator grant VH-NG-1248 Micro "Big Data."

### Author Contributions

JP Saraiva: data curation, software, formal analysis, validation, visualization, and writing—original draft, review, and editing.
A Bartholomäus: data curation, software, validation, visualization, methodology, and writing—review and editing.
R Kallies: resources, methodology, and writing—review and editing.
M Gomes: formal analysis, validation, and methodology.
M Bicalho: software and validation.
J Coelho Kasmanas: software, formal analysis, and writing—review and editing.
C Vogt: data curation and writing—original draft, review, and editing.
A Chatzinotas: resources and writing—review and editing.
P Stadler: methodology and writing—review and editing.
O Dias: conceptualization and writing—review and editing.
U Nunes da Rocha: conceptualization, resources, software, supervision, funding acquisition, investigation, methodology, project administration, and writing—original draft, review, and editing.

### Conflict of Interest Statement

The authors declare that they have no conflict of interest.

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
