## [Reviewer comments · Life Science Alliance]

Life Science Alliance

OrtSuite - from genomes to prediction of microbial interactions within targeted ecosystem processes

Joao Saraiva, Alexander Bartholomäus, René Kallies, Marta Gomes, Marcos Fleming Bicalho, Jonas Coelho Kasmanas, Carsten Vogt, Antonis Chatzinotas, Peter Stadler, Oscar Dias, and Ulisses Nunes da Rocha

DOI: <https://doi.org/10.26508/lisa.202101167>

Corresponding author(s): Ulisses Nunes da Rocha, Helmholtz Centre for Environmental Research

Review Timeline:

Submission Date:	2021-07-22
Editorial Decision:	2021-07-22
Revision Received:	2021-09-01
Editorial Decision:	2021-09-08
Revision Received:	2021-09-13
Accepted:	2021-09-14

Transaction Report:

Please note that the manuscript was previously reviewed at another journal and the reports were taken into account in the decision-making process at Life Science Alliance.

Reviewer #1 Review

Report for Author:

SUMMARY

In this manuscript entitled: "OrtSuite-form genomes to prediction of microbial interactions with targeted ecosystem processes", Saraiva et al., designed a bioinformatic workflow to predict possible microbial interactions within microbial communities considering the genetic content of the community and the required reactions/pathways to perform a specific biological process. Briefly, by providing the protein content of the studied microbial community as well as the protein sequences of the associated examined process along with a set of Gene-Protein-Reaction rules, the user can explore (i) whether individual species can perform the studied process (ie the species genetic

content covers the integrality of the protein requirements for the studied process), (ii) which combinations of species would perform the studied process and (iii) what would be the associated synergistic interactions. The authors' strategy to increase feasibility and decrease processing time was mainly to (i) focus on a given biological process (and not have to span the entire genome of every species) as well as (ii) use ortholog clustering for the community's species protein and (iii) screen only 50% of each cluster as a first step to identify clusters of proteins associated with the examined pathway. The workflow relies on 3 simple commands, the first one allowing the creation of the specific database containing all the information and sequences of the research pathway, the second one allowing the protein clustering of the studies microbial species and the 3rd one performing the functional annotation and the interaction inference. To test their approach, the authors studied the ability of different species (individually and as a community) to perform benzoate degradation. The vast majority of their analysis matched experimental data of an independent study and they were able to predict possible species interactions proposed in that previous experimental study as well.

GENERAL REMARKS

Altogether, Saraiva et al., are addressing a complex and very necessary problem in the research field of microbial community and microbial interactions and I believe their approach to be relevant for a hypothesis-driven inference of microbial interactions for specific biological process. However, I believe, that as it is, this manuscript may be too oriented towards and audience of bioinformaticians and would feel obscure for microbiologists with a limited background in bioinformatics. I believe that modifying the Introduction part by adding some more context/background and scope of utilization as well as reworking some of the Results section by adding more rationale information and methodological details would improve the clarity of the manuscript and broaden the targeted audience. Also, improving the quality of the Figures (especially Figure 1) would strongly improve the manuscript.

MAJOR POINTS

Below, are included several remarks for improvements, that I hope will be constructive:

A) Introduction

The introduction is pretty long and very technical. I believe that (i) adding more biological context and introducing why such an approach is needed from a broader perspective and not only in regards of the existing tools and their limitations, as well as (ii) describing more precisely and directly the scope of applications of this new tool (for example design of synthetic consortia) would facilitate the reader to grasp the relevance of this workflow.

B) Results

The first part of the Results section (down to the section entitled "Using Ort Suite to predict interspecies interactions") is complex to follow as it is lacking some rationale, clear conclusions, as well as essential methodological information to understand what the workflow does and why the authors are performing the different tests they are describing.

1- While the Methods part is remarkably clear about how the workflow works and what the different steps are (what they do and what tools they use), some of this information should be mentioned in the Results section too, so the reader starts with a clear and complete understanding of the workflow. The information provided in the first paragraph of the Results section is not enough and

quite confusing. This could also be helped by improving Figure 1, that should be a graphical explanation of the workflow (what it does and how). In its current form, the Figure remains unclear and disconnected from the referring text in the Results section. While the required information is clearly laid out in the Figure legend, that information is not conveyed in the figure itself. I believe the relevance and use of Figure1 could be improved by (i) graphically illustrating what is done rather than the tools used and (ii) clearly highlighting the 5 tasks that are performed by the workflow as mentioned from Lines 140 to 143. Maybe having a general figure to show the overall workflow and then sub-figures describing steps or tasks into more details could help.

2- Line 168, the authors mentioned they "performed an evaluation of the effects of point mutations during clustering". Can the authors explain more clearly the rationale behind that test, what information they are trying to obtain and why this is important for their workflow?

3- From Lines 206 to 211, the authors mentioned having used other annotation tools than KEGG. Could the authors explain why and describe more precisely the observed differences compared to KEGG beyond the simple processing time. And eventually, could the authors clearly state their conclusions and why they decided to use KEGG for the rest of the analysis.

4- Figure 3 is never mentioned in the Main Text.

C) Discussion

In the discussion section, could the authors also discuss more generally the limitations of their technique in understanding microbial interactions? That would help repositioning the relevance of their workflow in the general context of the study of microbial communities and broaden the perspectives of its applications.

1- Could the authors discuss whether the approach is limited to specific types of interactions such as cross-feeding, and specific processes such as product degradation and product synthesis. Have the authors used their workflow to study other communities, pathways and functions?

2- Could the authors also discussed that, while facilitating the analysis, implementing only a subset of target reactions, could lead to "false positive" interactions, as many other interactions are likely to interfere (for example antibiotic production inhibiting the growth of one species that could contribute to the targeted pathway).

3- Also, the authors mention that other reactions and parameters (such as transporters for instance) can be implemented to finetune the predictions. Have the authors also considered the use of transcriptomic and metatranscriptomic data that would highlight whether the genes are expressed when present, increasing the precision of the prediction? Maybe, the authors could discuss how this would be feasible in their approach?

Examples of limitations described in points 2 and 3 are briefly being alluded to in the discussion of the comparison of OrtSuite interactions predictions and the results obtained in the Fetzer study (Lines 364 to 384). I believe, this could be used to highlight more generally limitations and challenges in studying interactions.

MINOR POINTS

1- Lines 172 and 174 shouldn't that be BLAST instead of OrthoFinder?

2- Line 188. Possible typo. Should that be "the alignment to the sequences FROM ORAdb" ?

3- Line 551: Could the authors mentioned the name of the set?

Reviewer #2 Review

Report for Author:

The manuscript by Saraiva et al. presented a pipeline that aims to annotate genes and predict interactions between different microbial species. I like this idea very much, as most current annotation pipelines only focus on functional assignments. This manuscript is overall well written, even though there are some minor grammar mistakes, which can be easily solved by careful proof-reading. However, there are some large limitations for this manuscript and the pipeline itself.

1. Usage of specific software.

The software diamond now comes to v2 version. It is not clear which version is used in the pipeline. Based the citation, it seems to be v1. The authors should test v2 for better performances. Here is the newest citation: Buchfink B, Reuter K, Drost HG, "Sensitive protein alignments at tree-of-life scale using DIAMOND", Nature Methods 18, 366-368 (2021). doi:10.1038/s41592-021-01101-x

DIAMOND vs Blastp. I think most people accept that DIAMOND is faster and better than Blastp. So, I do not understand why the authors compared them and discussed this in very much detail in the manuscript. In fact, the authors should discuss another similar software, MMSeqs2. It is used in OrthoFinder, but not included in OrtSuite. I would recommend the authors to include the MMSeqs2 option. Also, I think there is no need to discuss which of these three tools are better, as they have been reported in other papers.

2. Functional annotation vs interaction predictions.

I found somehow these two were mixed in the manuscript. Of course, interaction predictions rely on good functional annotations. However, they are two different things. E.g. Metaerg only does functional annotations. This manuscript discussed a lot about the annotations but did not present too much about microbial interactions. It is the interactions that the authors claim to fame. The benzoate story is ok to illustrate the annotation aspect, while I find it is not suitable to explain microbial interactions. It is a very simple case. A case using benzene is possibly better. In the environment, microbial interactions are very complex. I expect microbial interactions should go as shown in this figure: <https://microbiomejournal.biomedcentral.com/articles/10.1186/s40168-017-0322-2/figures/6>.

The culture with mixed isolates is not a good example. I understand that the pipeline is designed for metagenome-assembled genomes, which are derived from environmental genomes. It should be normally a large dataset. Unfortunately, I did not see such an actual dataset was tested in the manuscript, from either environmental or host-associated samples, as the authors discussed in the introduction section. There are lots of metagenomic dataset with a larger number of reconstructed MAGs than the claimed 18.

3. Comparison with other pipelines.

This is clearly missing in the manuscript. For environmental samples, we tend to know more functional groups, while OrtSuite only focused on specific processes. In this perspective, this pipeline is not as useful as the combination of GhostKOALA or equivalents for functional assignments and KEGG Mapper for annotation visualization. Additionally, OrtSuite also needs manual inspection of the database. It is laborious. I also expect a comparison with other tools, e.g. METABOLIC and DRAM. They all have a taste for microbial interactions, even though they are not specifically designed for this aim.

4. Other missing information.

Figure 3 is not discussed in the manuscript. It is a good design for visualization and deserves more texts.

Although it was discussed in the manuscript, the tutorial was deleted in the Github webpage.

I would also recommend the authors to put all the required software dependencies into one conda package for easy installation.

Reviewer #3 Review

Report for Author:

The manuscript by Saraiva et al describes OrtSuite, a bioinformatics software to infer microbial interactions based on gene content analysis and orthologous groups predictions.

The idea of detecting synergistic organisms based on gene content and pathway profiling is indeed interesting. However, I do not think the method described has been sufficiently tested and described. First, the manuscript is too much focused on benchmarking the runtime of relatively simple and well studied workflows (e.g., orthology prediction and functional annotation) rather than on demonstrating the accuracy of OrtSuite in predicting interspecies interactions. Secondly, I think the results, as presented in the current version of the manuscript, do not support OrtSuite as a mature software and reliable method. In the following I provide some aspects that could be improved:

Although OrtSuite is presented as a generic prediction tool, only a case example is provided as a benchmark. Importantly, the sensitivity vs sensibility of the predictions are not really tested, reporting only the true positive rate from a single experiment. While case examples are always good as a validation, I think a more comprehensive and systematic benchmark should be presented.

A number of details about the methodology used to infer the microbial interactions are not clear. Most of the paper is about functional annotation and orthology inference, which is not the primary goal of this work and in fact relies on external software.

When it comes to inferring species interactions, a number of questions remain unclear: are full metabolic models being reconstructed for each genome? Are only partial metabolic routes from the different organisms being used to infer putative interactions? If so, what's the criterion used? Some software options are described in lines 499-519, but those options are not tested or further discussed in the results.

The section about recall rates in orthology prediction (lines 166-184) is confusing and probably

unnecessary. From the first lines, it seems authors are just testing OrthoFinder in Diamond vs Blast mode. However, the following lines refer to a comparison between OrthoFinder and Diamond (line 172-174). Most importantly, the performance tests presented are irrelevant for the manuscript and, in any case, already covered by the OrthoFinder original manuscript (e.g., BLAST vs DIAMOND runtime and sensitivity differences are known and previously described). Similarly, grouping artificially mutated sequences into the same orthologous group is an obvious result if no duplication events are simulated.

The section describing functional annotations of orthologous groups using different e-value thresholds does not seem to provide any significant insight. As the authors recognise in the text: "no striking difference was observed between the four different cutoffs". This is expected, as it is that runtime differences using different e-value thresholds is negligible.

Despite being claimed in the results section, I would not say the OrtSuite is a user-friendly tool in its current form. Installation is complex due to a number of dependencies (external software such as Diamond, OrthoFinder, pandoc, MCL, ...) and system requirements (Java, Python, R). I managed to install everything, but I don't really think the software is at the average user level. Additionally, output files are convoluted and hard to interpret. My suggestion would be to distribute the tool using any of the modern packaging systems (e.g., bioconda) and improve documentation.

July 22, 2021

Re: Life Science Alliance manuscript #LSA-2021-01167-T

Ulisses Nunes da Rocha
Helmholtz Centre for Environmental Research - UFZ
Germany

Dear Dr. Nunes da Rocha,

Thank you for submitting your manuscript entitled "Ort Suite - from genomes to prediction of microbial interactions within targeted ecosystem processes" to Life Science Alliance.

I saw your note that your postdoc is away. I am simply listing the requested revisions here so that you will be able to upload the revised files when they are ready.

- Address Reviewer 1's Major Points, which do not require additional experimentation
- Incorporate Reviewer 2's points into the Discussion, while removing emphasis on which tool is best
- Address Reviewer 3's questions regarding the inference of species interactions, and the point about describing Ort Suite as "user-friendly". Please also comment on the additional points in your Response to Reviewers.

Thank you for this interesting contribution to Life Science Alliance. We are looking forward to receiving your revised manuscript.

Sincerely,

Eric Sawey, PhD
Executive Editor
Life Science Alliance
<http://www.lsa-journal.org>

- A letter addressing the reviewers' comments point by point.
- An editable version of the final text (.DOC or .DOCX) is needed for copyediting (no PDFs).
- High-resolution figure, supplementary figure and video files uploaded as individual files: See our detailed guidelines for preparing your production-ready images, <https://www.life-science-alliance.org/authors>
- Summary blurb (enter in submission system): A short text summarizing in a single sentence the study (max. 200 characters including spaces). This text is used in conjunction with the titles of papers, hence should be informative and complementary to the title and running title. It should describe the context and significance of the findings for a general readership; it should be written in the present tense and refer to the work in the third person. Author names should not be mentioned.

B. MANUSCRIPT ORGANIZATION AND FORMATTING:

Dear Dr. Sawey,

We appreciate the comments of the reviewers who gave many interesting remarks and suggestions.

We believe that all comments have been considered in the revised version or addressed in the point-by-point reply. Briefly, we have added additional context of the use of OrtSuite to the introduction and improved the results section by adding the rationale of the work and more methodological details to reach a broader audience. We have also made changes to Figure 1 which describes the workflow employed in OrtSuite. Mainly, we clarified the different steps and tasks for better integration to the Methods and Results. We have also modified the method of installation of OrtSuite. We created two user-friendly methods for installation (via docker image or conda installation procedure) and instructions on how to use them. They can be found in the updated GitHub repository.

We hope the point-by-point replies are clear, and we are looking forward to your reply.

Sincerely,

Ulisses Nunes da Rocha

Reviewer #1:

SUMMARY

In this manuscript entitled: "OrtSuite-from genomes to prediction of microbial interactions with targeted ecosystem processes", Saraiva et al., designed a bioinformatic workflow to predict possible microbial interactions within microbial communities considering the genetic content of the community and the required reactions/pathways to perform a specific biological process. Briefly, by providing the protein content of the studied microbial community as well as the protein sequences of the associated examined process along with a set of Gene-Protein-Reaction rules, the user can explore (i) whether individual species can perform the studied process (ie the species genetic content covers the integrality of the protein requirements for the studied process), (ii) which combinations of species would perform the studied process and (iii) what would be the associated synergistic interactions. The authors' strategy to increase feasibility and decrease processing time was mainly to (i) focus on a given biological process (and not have to span the entire genome of every species) as well as (ii) use ortholog clustering for the community's species protein and (iii) screen only 50% of each cluster as a first step to identify clusters of proteins associated with the examined pathway. The workflow relies on 3 simple commands, the first one allowing the creation of the specific database containing all the information and sequences of the research pathway, the second one allowing the protein clustering of the studies microbial species and the 3rd one performing the functional annotation and the interaction inference. To test their approach, the authors studied the ability of different species (individually and as a community) to perform benzoate degradation. The vast majority of their analysis matched experimental data of an independent study and they were able to predict possible species interactions proposed in that previous experimental study as well.

GENERAL REMARKS

Altogether, Saraiva et al., are addressing a complex and very necessary problem in the research field of microbial community and microbial interactions and I believe their approach to be relevant for a hypothesis-driven inference of microbial interactions for specific biological process. However, I believe that as it is, this manuscript may be too oriented towards an audience of bioinformaticians and would feel obscure for microbiologists with a limited background in bioinformatics. I believe that modifying the Introduction part by adding some more context/background and scope of utilization as well as reworking some of the Results section by adding more rationale information and methodological details would improve the clarity of the manuscript and broaden the targeted audience. Also, improving the quality of the Figures (especially Figure 1) would strongly improve the manuscript.

Reply 1: We thank the reviewer for his comprehensive comments on the manuscript. In the general remarks of reviewer 1, we identified four main issues.

- **1.** The first issue is related to the “excess” orientation of the manuscript towards bioinformaticians. We do agree that running tools for functional annotation of genomes and prediction of microbial interactions will always require users to possess basic skills in bioinformatics. We do, however, facilitate the use of OrtSuite by non-bioinformaticians by providing easy to follow installation and execution of the tool, described in the manuscript, Lines 153-179. This has been improved by including a docker installation and conda installation (for installation in High Performance Computers -HPCs) as well as a detailed guide in the github repository (<https://github.com/mdsufz/OrtSuite>).
- **2.** The second issue we identified is adding context and background to the introduction as well as scope of utilization. We addressed this issue by adding text to the introduction highlighting the potential of OrtSuite to identify key species involved in ecosystem processes (Lines 124-126). Further, we also mention how the identification of synergistic species interactions can lead to the design of synthetic microbial communities that can be used in processes such as bioremediation, energy production and human health (Lines 126-129).
- **3.** The third issue is related to the lack of clarity in the Results section. Methodology details and rationale were missing/incomplete. In order to improve clarity of the methods and rationale behind OrtSuite, we made several changes to the Results section. An introductory paragraph was added briefly explaining the motivation behind OrtSuite as well as the process of integrating targeted functional annotation with prediction of synergistic species interactions (Lines 146-150). Also, the text in the first subsection (“OrtSuite is a flexible and user-friendly pipeline“, lines 152) was modified providing a more detailed description of the different steps and requirements in OrtSuite. These include the requirement of only a text file, provided by the user, with the list of identifiers used to populate the custom database and a brief explanation as well as the inputs and outputs required and generated during each step of the pipeline.
- **4.** The fourth issue we identified is the improvement of Figure 1. We modified Figure 1 by including the numbering of the different steps that compose each task in OrtSuite. Further, the background colouring is meant to clearly distinguish the three tasks.

MAJOR POINTS

Below, are included several remarks for improvements, that I hope will be constructive:

A) Introduction

The introduction is pretty long and very technical. I believe that (i) adding more biological context and introducing why such an approach is needed from a broader perspective and not only in regards of the existing tools and their limitations, as well as (ii) describing more precisely and directly the scope of applications of this new tool (for example design of synthetic consortia) would facilitate the reader to grasp the relevance of this workflow.

Reply 2: We would like to thank the reviewer for his constructive suggestions. The two main issues in this comment are the inclusion of more biological context (e.g. how do the results in OrtSuite help resolve biological questions) and a more precise description of the scope of application of Ortsuite. We addressed both issues by including text in the introduction to provide more biological context to the usefulness of predicting synergistic interspecies interactions (Lines 124-126). We highlight the potential of OrtSuite to identify key species involved in ecosystem processes as well as the use of the results in the design of synthetic microbial communities with applications in bioremediation, energy production and human health (Lines 126-129).

B) Results

The first part of the Results section (down to the section entitled "Using OrtSuite to predict interspecies interactions") is complex to follow as it is lacking some rationale, clear conclusions, as well as essential methodological information to understand what the workflow does and why the authors are performing the different tests they are describing.

Reply 3: The general remark in point B of the reviewer points to two main issues.

- **1.** The first issue we identify is the lack of rationale in the results section. We have addressed this issue by adding an introductory paragraph briefly explaining the motivation behind OrtSuite as well as the process of integrating targeted functional annotation with prediction of synergistic species interactions (Lines 146-150).
- **2.** The second issue we identified is the lack of essential methodological information in the results section that can provide the reader with a better understanding of the different steps in OrtSuite. We have addressed this issue by modifying the text in the first subsection ("OrtSuite is a flexible and user-friendly pipeline") where we provide a more detailed description of the different steps and requirements in OrtSuite (Lines 153-179). We further describe the requirement of only a text file, provided by the user, with the list of identifiers used to populate the custom database and a brief explanation as well as the inputs and outputs required and generated during each step of the pipeline (Lines 153-179).

1- While the Methods part is remarkably clear about how the workflow works and what the different steps are (what they do and what tools they use), some of this information should be mentioned in the Results section too, so the reader starts with a clear and complete understanding of the workflow. The information provided in the first paragraph of the Results section is not enough and quite confusing. This could also be helped by improving Figure 1, that should be a graphical explanation of the workflow (what it does and how). In its current form, the Figure remains unclear and disconnected from the referring text in the Results section. While the required information is clearly laid out in the Figure legend, that information is not conveyed in the figure itself. I believe the relevance and use of Figure 1 could be improved by (i) graphically illustrating what is done rather than the tools used and (ii) clearly highlighting the 5 tasks that are performed by the workflow as mentioned from Lines 140 to 143. Maybe having a general figure to show the overall workflow and then sub-figures describing steps or tasks into more details could help.

Reply 4: We thank the reviewer for his suggestions. In this comment we identified two main issues that are interconnected: the lack of a clear understanding of the workflow and disconnection of Figure 1 to the results section.

- The first issue was addressed by the inclusion of an introductory paragraph briefly explaining the motivation behind OrtSuite as well as the process of integrating targeted functional annotation with prediction of synergistic species interactions (Lines 146-150). Further, the text in the first subsection of the results ("OrtSuite is a flexible and user-friendly pipeline") was modified to provide a more detailed description of the different steps and requirements in OrtSuite. Figure 1 was also modified to reflect the text in the manuscript. Modifications included the naming of the different steps involved in each task (Lines 159-167). To improve the clarity of the Figure 1, we made clear the pipeline is divided into 3 tasks (scripts) divided in 5 steps. Task 1 and 3 are divided into 2 steps, while task 2 consists of one step. These changes were also described in Figure 1's legend (Lines 864-865).

2- Line 168, the authors mentioned they "performed an evaluation of the effects of point mutations during clustering". Can the authors explain more clearly the rationale behind that test, what information they are trying to obtain and why this is important for their workflow?

Reply 5: We thank the reviewer for his constructive comments. Point mutations can alter the function of genes by altering amino acid composition. Thus, we performed a simple test to determine if this was the case for point mutation rates up to 25% (Lines 627-638). Further, we have included text to emphasize the potential effects in the clustering of orthologs in the presence of point mutations is Lines 191-192.

3- From Lines 206 to 211, the authors mentioned having used other annotation tools than KEGG. Could the authors explain why and describe more precisely the observed differences compared to KEGG beyond the simple processing time. And eventually, could the authors clearly state their conclusions and why they decided to use KEGG for the rest of the analysis.

Reply 6: We thank the reviewer for the comment. We included other annotation tools to highlight the fact that they perform full genome annotations which would require additional manual processing to compare results. Since OrtSuite already was able to match 96% of KEGG annotations, the additional file

processing would not be cost-effective. We have now included text in lines 238-240 to reflect the laborious process of filtering pathways of interest and predicting interspecies interactions.

4- Figure 3 is never mentioned in the Main Text.

Reply 7: We thank the reviewer for the comment. Indeed, the mentioning of Figure 3 was mistakenly absent from its intended places. We have now added reference to Figure 3 in lines 357 and 552.

C) Discussion

In the discussion section, could the authors also discuss more generally the limitations of their technique in understanding microbial interactions? That would help repositioning the relevance of their workflow in the general context of the study of microbial communities and broaden the perspectives of its applications.

Reply 8: In the general comment of Reviewer 1 to the Discussion we identified the issue of the lack of a general description of the limitations of OrtSuite in understanding microbial interactions. In lines 546-547 of the manuscript we state that OrtSuite only allows to infer synergistic interactions such as cross-feeding. We have, however, added text to the manuscript stating that the inclusion of other omics data, such as transcriptomics could improve our understanding of microbial interactions by mapping functional potential to gene expression. We do state that this integration is not trivial and would increase complexity to the problem and is out of scope of OrtSuite. The added text can be found in Lines 418-422.

1- Could the authors discuss whether the approach is limited to specific types of interactions such as cross-feeding, and specific processes such as product degradation and product synthesis. Have the authors used their workflow to study other communities, pathways and functions?

Reply 9: We thank the reviewer for his comments. In this comment we identified two main issues: The use of OrtSuite to study other types of interactions and its application to other ecosystem processes. In OrtSuite, prediction of interactions is based on the complementarity of genomic content from different species to perform a complete set of reactions and, thus, only synergistic species interactions are considered such as cross-feeding or due to the removal of toxic compounds from the environment. However, users do have the possibility to filter putative interactions based on prior knowledge or biological feasibility. This is stated in lines 545-547. In regards to processes, OrtSuite is not limited to production degradation or synthesis. It does, however, require that the gene requirements for each reaction involved in a process be known as stated in lines 455 and 457 of the manuscript. In our study, we tested benzoate because the molecular mechanisms of transformation (aerobically and anaerobically) are well described and data from an independent study was available, which we used to validate our analysis.

2- Could the authors also discussed that, while facilitating the analysis, implementing only a subset of target reactions, could lead to "false positive" interactions, as many other interactions are likely to interfere (for example antibiotic production inhibiting the growth of one species that could contribute to the targeted pathway).

Reply 10: We thank the reviewer for his comments. OrtSuite predicts putative synergistic interactions based on the combined genomic content of species focused only on a specific set of reactions. In the example provided in the manuscript, antibiotic resistance was not included. However, should the users want to include such complementary pathways or sets of reactions they can do so by adding those reactions to the list that is used for populating the database (Lines 424-427) . Nevertheless, manual inspection and filtering of predictions will always be required to decrease the number of false positives and reflect current knowledge. We have added text to highlight this fact in the manuscript in Lines 172-174.

3- Also, the authors mention that other reactions and parameters (such as transporters for instance) can be implemented to finetune the predictions. Have the authors also considered the use of transcriptomic and metatranscriptomic data that would highlight whether the genes are expressed when present, increasing the precision of the prediction? Maybe, the authors could discuss how this would be feasible in their approach?

Reply 11: We thank the reviewer for his comments. Indeed the addition of other omics data can improve predictions of interspecies interactions. We have added text to the manuscript in Lines 418-422, where we discuss the use and potential integration of (meta)transcriptomic data to refine synergistic species interactions by limiting predictions based on gene expression levels. We further mention the increase in complexity when integrating this type of data.

Examples of limitations described in points 2 and 3 are briefly being alluded to in the discussion of the comparison of OrtSuite interactions predictions and the results obtained in the Fetzer study (Lines 364 to 384). I believe, this could be used to highlight more generally limitations and challenges in studying interactions.

Reply 12: We thank the reviewer for his comments. We have added text to the manuscript (Lines 455-460) highlighting the limitations of OrtSuite and the challenges in predicting microbial interactions. We also mention the need for integration with other omics data to improve predictions.

MINOR POINTS

1- Lines 172 and 174 shouldn't that be BLAST instead of OrthoFinder?

Reply 13: We thank the reviewer for the comment. In the manuscript the comparison should be between BLAST and Diamond and not between OrthoFinder and Diamond. This error has been corrected (Line 201 and Line 202).

2- Line 188. Possible typo. Should that be "the alignment to the sequences FROM ORAdb"?

Reply 14: We thank the reviewer for the comment. Yes, the word "from" is missing. We have now added it (Line 216).

3- Line 551: Could the authors mentioned the name of the set?

Reply 15: We thank the reviewer for the comment. We have now included the names of the species with artificially mutated genomes (Lines 592-593).

Reviewer #2:

The manuscript by Saraiva et al. presented a pipeline that aims to annotate genes and predict interactions between different microbial species. I like this idea very much, as most current annotation pipelines only focus on functional assignments. This manuscript is overall well written, even though there are some minor grammar mistakes, which can be easily solved by careful proof-reading. However, there are some large limitations for this manuscript and the pipeline itself.

1. Usage of specific software.

The software diamond now comes to v2 version. It is not clear which version is used in the pipeline. Based the citation, it seems to be v1. The authors should test v2 for better performances. Here is the newest citation: Buchfink B, Reuter K, Drost HG, "Sensitive protein alignments at tree-of-life scale using DIAMOND", Nature Methods 18, 366-368 (2021). doi:10.1038/s41592-021-01101-x

Reply 16: We thank the reviewer for the comments. Indeed the version of DIAMOND used in OrtSuite is v1 (more precisely v0.9.22). We have now added the version of DIAMOND to the manuscript in line 513. According to the authors of the manuscript indicated by the reviewer, the new version of DIAMOND now includes two additional sensitivity modes (very sensitive and ultra sensitive). However, the authors claim that the improvements mostly occur at the computational speedup level. In our study, the use of DIAMOND in Step 4 (Functional annotation) only took 6 minutes to complete which is already a quick procedure and thus no significance gains are expected.

DIAMOND vs Blastp. I think most people accept that DIAMOND is faster and better than Blastp. So, I do not understand why the authors compared them and discussed this in very much detail in the manuscript. In fact, the authors should discuss another similar software, MMSeqs2. It is used in OrthoFinder, but not included in OrtSuite. I would recommend the authors to include the MMSeqs2 option. Also, I think there is no need to discuss which of these three tools are better, as they have been reported in other papers.

Reply 17: We thank the reviewer for the comments. The issues we identified in this comment is the detailed description of the comparison between BLASTp and DIAMOND during the clustering of orthologs and the absence of MMSeqs in the same tests. The use of BLASTp or DIAMOND has gathered much debate, with the main issues revolving around the selection of the less-sensitive and fast DIAMOND or the more sensitive and slower BLASTp. The recent publication of DIAMOND, as mentioned by the reviewer, describes that DIAMOND is now capable of obtaining accuracy levels comparable to BLASTp whilst significantly decreasing the time needed to perform sequence alignments. At the time of the development of OrtSuite this work had not yet been published and thus, we decided to perform this simple analysis. The unavailability of MMSeqs2 in OrthoFinder when OrtSuite was developed coupled with the fact that DIAMOND and BLAST are the most commonly used sequence aligners, steered us to

test only the latter two. Nevertheless, we have added the option of MMSeqs2 for OrthoFinder in OrtSuite. The text in the manuscript that clarifies these points, as well as in the availability of MMSeqs2 option in OrthoFinder, can be found in Lines 194-197 and 512. Additionally, we have updated the github repository with the information required for users to select MMSeqs2 as the sequence aligner.

2. Functional annotation vs interaction predictions.

I found somehow these two were mixed in the manuscript. Of course, interaction predictions rely on good functional annotations. However, they are two different things. E.g. Metaerg only does functional annotations. This manuscript discussed a lot about the annotations but did not present too much about microbial interactions. It is the interactions that the authors claim to fame. The benzoate story is ok to illustrate the annotation aspect, while I find it is not suitable to explain microbial interactions. It is a very simple case. A case using benzene is possibly better. In the environment, microbial interactions are very complex. I expect microbial interactions should go as shown in this figure:

<https://microbiomejournal.biomedcentral.com/articles/10.1186/s40168-017-0322-2/figures/6>.

The culture with mixed isolates is not a good example. I understand that the pipeline is designed for metagenome-assembled genomes, which are derived from environmental genomes. It should be normally a large dataset. Unfortunately, I did not see such an actual dataset was tested in the manuscript, from either environmental or host-associated samples, as the authors discussed in the introduction section. There are lots of metagenomic dataset with a larger number of reconstructed MAGs than the claimed 18.

Reply 18: We thank the reviewer for his comments. In this comment we identified the following issues: the limited discussion of microbial interactions and the use of cultures with mixed isolates to predict microbial interactions. OrtSuite predicts synergistic species interactions based on their functional potential and their complementarity to perform a given set of reactions. As the reviewer correctly states in his comments, annotation and prediction of interactions are tightly connected and, thus, mostly intertwined throughout the discussion. The use of the data from the Fetzer study was to assess whether the predicted species interactions by the authors could be substantiated by the species' functional potential. As the reviewer comments, with the advances in sequencing technologies comes the increase in the number of genomes recovered from metagenomes. However, and as stated in the manuscript in Lines 430-442, the average number of MAGs, per library, from three major studies did not exceed 16. Thus, predicting annotation and species interactions in a single putative sample consisting of 18 MAGs was not unrealistic. Further, we relied on the knowledge of experts (Dr. Carsten Vogt) to accurately characterize subsections of benzoate degradation. One of the advantages of OrtSuite is that users can predict interspecies interactions in reaction subsets.

3. Comparison with other pipelines.

This is clearly missing in the manuscript. For environmental samples, we tend to know more functional groups, while OrtSuite only focused on specific processes. In this perspective, this pipeline is not as useful as the combination of GhostKOALA or equivalents for functional assignments and KEGG Mapper

for annotation visualization. Additionally, OrtSuite also needs manual inspection of the database. It is laborious. I also expect a comparison with other tools, e.g. METABOLIC and DRAM. They all have a taste for microbial interactions, even though they are not specifically designed for this aim.

Reply 19: We thank the reviewer for the comments. In this comment, we identified the issues of OrtSuite not being as useful as GhostKOALA and KEGG Mapper for annotation and visualization, respectively. Further, we also identified the issue of manual inspection of the database and lack of comparison to other tools.

GhostKOALA and KEGG Mapper are good tools for complete genome functional annotation and visualization. However, a substantial manual processing is required to extract specific ecosystem processes. The idea behind OrtSuite is to precisely avoid performing full genome-scale annotation. Further, identifying putative interspecies interactions will require substantial manual work as users will need to scan all results to extract the genes of interest from each genome and assess which species complement each other, metabolically, to perform the process of interest. OrtSuite facilitates the entire process by only performing targeted functional annotation and automatically calculating combinations of species whose functional profiles could complement each other to perform a set of reactions of interest. Given that METABOLIC and DRAM are not designed for identification of species interactions we did not perform any comparison tests with these.

In lines 172-174 of the manuscript we recommend the inspection of the database and Gene-Protein-Reactions rules after steps 1 and 2 (ORA database generation and Gene-Protein-Reaction (GPR) rules) due to the constant addition of new entries and updates. Nevertheless, we have included text in line 174, reinforcing that manual inspection is not mandatory.

4. Other missing information.

Figure 3 is not discussed in the manuscript. It is a good design for visualization and deserves more texts.

Reply 20: We thank the reviewer for the comment. Indeed, the mentioning of Figure 3 was mistakenly absent from its intended places. We have now added this in lines 357 and 552.

Although it was discussed in the manuscript, the tutorial was deleted in the Github webpage.

Reply 21: We thank the reviewer for the comment. We apologize but the sentence referring to this tutorial was mistakenly added and is now removed.

I would also recommend the authors to put all the required software dependencies into one conda package for easy installation.

Reply 22: We thank the reviewer for the comments. The issue we identified in this comment was the difficulty in installing all software dependencies to run OrtSuite. We have modified the method of installation of OrtSuite to include two methods of installation available to users: via docker or via conda. We recommend the installation of OrtSuite via docker, however, in certain systems such as HPCs, dockers are occasionally not allowed. In these cases, we provide installation guidelines via conda which, unfortunately, will require the individual installation of OrtSuite's dependencies. The option of docker and conda as installation methods is stated in lines 175-179 of the manuscript. Additionally, the github repository has also been updated to reflect these changes.

Reviewer #3:

The manuscript by Saraiva et al describes OrtSuite, a bioinformatics software to infer microbial interactions based on gene content analysis and orthologous groups predictions.

The idea of detecting synergistic organisms based on gene content and pathway profiling is indeed interesting. However, I do not think the method described has been sufficiently tested and described. First, the manuscript is too much focused on benchmarking the runtime of relatively simple and well studied workflows (e.g., orthology prediction and functional annotation) rather than on demonstrating the accuracy of OrtSuite in predicting interspecies interactions. Secondly, I think the results, as presented in the current version of the manuscript, do not support OrtSuite as a mature software and reliable method. In the following I provide some aspects that could be improved:

Reply 23: We thank the reviewer for his comments. In this comment, we identified the issue related to the benchmarking of OrtSuite using only one experiment. In our study, we tested benzoate because the molecular mechanisms of transformation (aerobically and anaerobically) are well described and data from an independent study was available, which we used to validate our analysis.

Although OrtSuite is presented as a generic prediction tool, only a case example is provided as a benchmark. Importantly, the sensitivity vs sensibility of the predictions are not really tested, reporting only the true positive rate from a single experiment. While case examples are always good as a validation, I think a more comprehensive and systematic benchmark should be presented.

Reply 24: We highly appreciate the suggestion for the calculation of sensitivity and sensibility. When developing OrtSuite we considered calculating sensitivity and specificity. However, this would require the knowledge of false positives and false negatives. Unfortunately, the false positives and false negatives cannot be obtained easily. It is nearly impossible to prove that there is no interaction (true negative) because the interaction could be very minor or substances could be shared in tiny amounts. Thus, we think that sensitivity and specificity cannot be calculated with the available data. Therefore, we decided not to add further discussion about this issue in the manuscript.

A number of details about the methodology used to infer the microbial interactions are not clear. Most of the paper is about functional annotation and orthology inference, which is not the primary goal of

this work and in fact relies on external software.

When it comes to inferring species interactions, a number of questions remain unclear: are full metabolic models being reconstructed for each genome? Are only partial metabolic routes from the different organisms being used to infer putative interactions? If so, what's the criterion used? Some software options are described in lines 499-519, but those options are not tested or further discussed in the results.

Reply 25: We thank the reviewer for the comment. In this comment the issue is describing in clearer terms how species interactions are inferred. In OrtSuite, annotation of the genomes is only performed based on the sequences present in ORAdb, thus reflecting only partial annotation. This process is stated in the manuscript in lines 523-533. Putative species interactions are based on their combined genomic potential to perform the ecosystem process of interest which can be further filtered based on Gene-Protein-Reaction rules and transporter-associated genes, among others. This information is presented in the manuscript in lines 536-552.

The section about recall rates in orthology prediction (lines 166-184) is confusing and probably unnecessary. From the first lines, it seems authors are just testing OrthoFinder in Diamond vs Blast mode. However, the following lines refer to a comparison between OrthoFinder and Diamond (line 172-174). Most importantly, the performance tests presented are irrelevant for the manuscript and, in any case, already covered by the OrthoFinder original manuscript (e.g., BLAST vs DIAMOND runtime and sensitivity differences are known and previously described). Similarly, grouping artificially mutated sequences into the same orthologous group is an obvious result if no duplication events are simulated.

Reply 26: We thank the reviewer for the comment. In this comment we identified the issue of testing performance of DIAMOND versus BLASTp during the clustering of orthologs. Additionally, some confusion exists since text in the manuscript says OrthoFinder versus DIAMOND. Indeed, in the manuscript the comparison should be between BLAST and Diamond and not between OrthoFinder and Diamond. This error has been corrected and is shown in Lines 201-202. The use of BLASTp or DIAMOND has gathered much debate with the main issues revolving around the selection of the less-sensitive and fast DIAMOND or the more sensitive and slower BLASTp. The most recent publication of DIAMOND (Buchfink B, Reuter K, Drost HG, "Sensitive protein alignments at tree-of-life scale using DIAMOND", Nature Methods 18, 366-368 (2021). doi:10.1038/s41592-021-01101-x), states that DIAMOND is now capable of obtaining accuracy levels comparable to BLASTp whilst significantly decreasing the time needed to perform sequence alignments. Nevertheless, the tests performed in our study only support the now published results of the new DIAMOND version. In regards to the testing of effects of point mutations during clustering of orthologs: while OrthoFinder's original manuscript does perform comprehensive performance tests it did not evaluate the effects of point mutations in the clustering of orthologs, hence the inclusion of these results in our manuscript. Furthermore, literature has described the potential for drastic effects in protein function even with changes at single base level which was the main rationale for performing these tests. This is described in the manuscript in lines 335-339.

The section describing functional annotations of orthologous groups using different e-value thresholds

does not seem to provide any significant insight. As the authors recognise in the text: "no striking difference was observed between the four different cutoffs". This is expected, as it is that runtime differences using different e-value thresholds is negligible.

Reply 27: We thank the reviewer for the comment. The issue we identify in this comment is the testing of different e-value cutoffs during functional annotation. In our study, we provide users with the ability to tune sequence alignment cutoffs. However, our aim in running these tests was to determine which default cutoff could be used while maintaining accurate results and not specifically focused on runtimes. This is mentioned in the manuscript in the results section "High rate of KEGG annotations predicted by OrtSuite" (Line 214). While no significant difference in annotation is observed between the different cutoffs, a larger drop in the number of clusters that transition from the relaxed search to the restrictive occurs when using an e-value cutoff of $1e^{-16}$.

Despite being claimed in the results section, I would not say the OrtSuite is a user-friendly tool in its current form. Installation is complex due to a number of dependencies (external software such as Diamond, OrthoFinder, pandoc, MCL, ...) and system requirements (Java, Python, R). I managed to install everything, but I don't really think the software is at the average user level. Additionally, output files are convoluted and hard to interpret. My suggestion would be to distribute the tool using any of the modern packaging systems (e.g., bioconda) and improve documentation.

Reply 28: We thank the reviewer for the comments. The issue we identified in this comment was the difficulty in installing all software dependencies to run OrtSuite. We have modified the method of installation of OrtSuite to include two methods of installation available to users: via docker or via conda. We recommend the installation of OrtSuite via docker. However, in certain systems such as HPCs, dockers are occasionally not allowed. In these cases, we provide installation guidelines via conda which, unfortunately, will require the individual installation of OrtSuite's dependencies. The option of docker and conda as installation methods is stated in lines 175-179 of the manuscript. Additionally, the github repository has also been updated to reflect these changes.

September 8, 2021

RE: Life Science Alliance Manuscript #LSA-2021-01167-TR

Dr. Ulisses Nunes da Rocha
Helmholtz Centre for Environmental Research - UFZ
Environmental Microbiology
Permoserstraße 15
Leipzig 04318
Germany

Dear Dr. Nunes da Rocha,

Thank you for submitting your revised manuscript entitled "OrtSuite - from genomes to prediction of microbial interactions within targeted ecosystem processes". We would be happy to publish your paper in Life Science Alliance pending final revisions necessary to meet our formatting guidelines.

- please add the Twitter handle of your host institute/organization as well as your own or/and one of the authors in our system
- please be sure that all Authors are listed in the Authors Contribution Section in the manuscript text
- please add callouts for Figure 3A-C to your main manuscript text
- please upload all your Tables in editable .doc or excel format

LSA now encourages authors to provide a 30-60 second video where the study is briefly explained. We will use these videos on social media to promote the published paper and the presenting author. Corresponding or first-authors are welcome to submit the video. Please submit only one video per manuscript. The video can be emailed to contact@life-science-alliance.org

A. FINAL FILES:

B. MANUSCRIPT ORGANIZATION AND FORMATTING:

Sincerely,

September 14, 2021

RE: Life Science Alliance Manuscript #LSA-2021-01167-TRR

Dr. Ulisses Nunes da Rocha
Helmholtz Centre for Environmental Research
Environmental Microbiology
Permoserstraße 15
Leipzig 04318
Germany

Dear Dr. Nunes da Rocha,

Thank you for submitting your Resource entitled "Ort Suite - from genomes to prediction of microbial interactions within targeted ecosystem processes". It is a pleasure to let you know that your manuscript is now accepted for publication in Life Science Alliance. Congratulations on this interesting work.

DISTRIBUTION OF MATERIALS:

Again, congratulations on a very nice paper. I hope you found the review process to be constructive and are pleased with how the manuscript was handled editorially. We look forward to future exciting submissions from your lab.

Sincerely,
